# An idiosyncratic zonated stroma encapsulates desmoplastic liver metastases and originates from injured liver

Carlos Fernández Moro [1,2,3,14], Natalie Geyer[1,14], Sara Harrizi[1,14], Yousra Hamidi[1,14], Sara Söderqvist[1], Danyil Kuznyecov [4], Evelina Tidholm Qvist[2], Media Salmonson Schaad[1], Laura Hermann[1], Amanda Lindberg[5], Rainer L. Heuchel [6], Alfonso Martín-Bernabé[7], Soniya Dhanjal[8], Anna C. Navis [8], Christina Villard[9], Andrea C. del Valle [1], Lorand Bozóky[1], Ernesto Sparrelid[10], Luc Dirix [11], Carina Strell[5,12], Arne Östman[7], Bernhard Schmierer [8], Peter B. Vermeulen[11], Jennie Engstrand [10], Béla Bozóky[2] & Marco Gerling [1,13] ✉

A perimetastatic capsule is a strong positive prognostic factor in liver metastases, but its origin remains unclear. Here, we systematically quantify the capsule's extent and cellular composition in 263 patients with colorectal cancer liver metastases to investigate its clinical significance and origin. We show that survival improves proportionally with increasing encapsulation and decreasing tumor-hepatocyte contact. Immunostaining reveals the gradual zonation of the capsule, transitioning from benign-like NGFR^high stroma at the liver edge to FAP^high stroma towards the tumor. Encapsulation correlates with decreased tumor viability and preoperative chemotherapy. In mice, chemotherapy and tumor cell ablation induce capsule formation. Our results suggest that encapsulation develops where tumor invasion into the liver plates stalls, representing a reparative process rather than tumor-induced desmoplasia. We propose a model of metastases growth, where the efficient tumor colonization of the liver parenchyma and a reparative liver injury reaction are opposing determinants of metastasis aggressiveness.

Tumor cell invasion into healthy tissue is a hallmark of cancer[1]. Metastatic tumor cells utilize different anatomic trajectories to colonize healthy organs. For instance, in liver metastases of colorectal cancer, tumor cells frequently invade the hepatic plates by replacing hepatocytes and co-opting the stromal scaffolds[2]. Melanoma metastases can exploit intra- and perivascular spaces for invasion, a trait rarely observed in other tumor types[2,3]. These tumor-specific patterns emphasize the role of distinct capabilities for metastatic invasion. However, one growth pattern is shared by liver metastases of different primary tumors and is uniformly associated with a favorable prognosis: Metastases encapsulated by a rim of fibrotic stroma have a more

favorable outcome in colorectal, gastric, and pancreatic adenocarcinomas, as well as in breast cancer and melanoma[2,4–9].

Clinically, the distinct patterns of invasion are best studied in colorectal cancer liver metastases (CRLM) because surgical resection is the standard-of-care, resulting in abundant specimens[10]. The two major CRLM growth patterns have been systematized into the so-called *desmoplastic* and *replacement* patterns[2]. In this nomenclature, "desmoplastic" refers to the capsule of extracellular matrix, immune cells, and fibroblast-like cells that separates tumor and liver[2,7,11]. In contrast, the "replacement" pattern is defined by direct contact between the hepatocytes and the invading tumor cells[2,7,12]. Numerous retrospective

studies have documented a survival benefit for CRLM patients with predominantly encapsulated/desmoplastic metastases[4,13,14].

However, the molecular bases for the emergence of the perimetastatic capsule remains unknown[2]. No associations between the growth patterns and oncogenic *KRAS*, *BRAF*, and *NRAS* mutations, which characterize aggressive colorectal cancer subtypes, have been found[14,15]. Microsatellite instability (MSI) is more frequent in desmoplastic than in replacement-type metastases; however, its absolute frequencies are low in both[14], and hence, MSI does not sufficiently explain growth pattern biology.

Desmoplasia is typically considered the result of resident fibroblast activation, driven by tumoral factors and reinforced by the recruitment of other cellular sources; in the prevailing model, desmoplasia is seen either as a damage response of the stroma to the presence of tumor cells or as a result of tumor cell-derived factors actively shaping their stromal niche[16].

Here, we sought a deeper understanding of the clinical implications and the origin of the perimetastatic capsule by analyzing CRLM growth patterns at high resolution using extensive digital annotations and multiplex in situ stains. Our results reveal a reparative hepatic process at the liver edge of the capsule, associated with impaired tumor viability. In liver metastases mouse models, we find that chemotherapy and tumor cell ablation induce encapsulation. Together, our data provide evidence that impaired tumor invasion in combination with a reparative injury response, rather than active induction by metastatic cells, drives capsule formation.

## Results

### Extended sampling and growth pattern annotations reveal fine-grained prognostic stratification

Conventionally, growth pattern fractions are estimated by visual assessments of the tumor-liver interface on hematoxylin & eosin-stained sections (Fig. 1a)[2]. While the international guidelines suggest the term "desmoplastic" for fibrotic areas separating tumor cells from the liver parenchyma[2], we here use "encapsulated", which is neutral to the origin of the stroma. Based on the conventional scoring approach, the current guidelines recommend a cut-off of 100% encapsulation for prognostic stratification[2], which was derived from testing different strata of encapsulation in large patient cohorts[2,14,17]. This cut-off implies a deterministic Boolean model, where only those metastases that are fully encapsulated have a significantly better prognosis compared to those with any fraction of non-encapsulated growth[14,17]. We reasoned that such a model is difficult to reconcile with the probabilistic character of biological systems, which would predict survival to increase with the extent of a positive physical trait, in this case, the proportion of encapsulation.

Conventional clinical scoring has two main shortcomings: (1) the limited sampling extent, leading to an incomplete representation of the tumor-liver interface, and (2), the scoring resolution, which is a visual estimate rather than a measurement. Both could explain the lack of prognostic resolution in tumors with growth pattern heterogeneity. To test the prognostic value of growth pattern scoring after addressing these caveats, we built a retrospective cohort of CRLM patients in which the growth patterns were systematically measured. First, we extensively sampled the invasion front of all metastases for each patient, approximating a panoramic central slice along the largest diameter of each metastasis (Fig. 1b and Supplementary Fig. 1)[18]. Next, we used digital whole slide images (WSIs) of all resulting sections to annotate the growth patterns over the entire invasion front (Fig. 1b). Our cohort comprised all consecutive patients undergoing their first operation for CRLM between May 2012 and December 2015 at Karolinska University Hospital (n = 263 patients, Table 1). This cohort reflects most modern CRLM therapies while allowing for a long follow-up time. The cohort comprised more male than female patients (163

males, 100 females), which corresponds well to the distribution previously reported[2].

For the first n = 94 patients, we assessed the growth patterns both visually and with our WSI-based approach. As expected, scores from digital annotations were more granular, while visual scoring was often performed in 5–10% increments (Supplementary Fig. 2a). Overall, we found good agreement between annotations and visual assessments performed by the same raters with more than one year between the assessments (ET, DK, Cohen's Kappa [predominant pattern] = 0.82) or when compared to external expert assessment (PV, who coined the patterns[12]; Cohen's Kappa [predominant pattern] = 0.72). Visual estimates deviated from annotation-based scoring, particularly but not exclusively towards 0% and 100% (Supplementary Fig. 2b). Next, we assessed whether visual estimates and digital annotations changed the classification of each patient into the categories of their predominant pattern. The classification remained unchanged for most patients, while n = 5 patients (5%) were reclassified (Fig. 1c). However, using different cut-offs for encapsulation as previously suggested (<33%, 33–<100%, and 100% desmoplastic pattern)[17], n = 18 patients (19%) were reclassified (Fig. 1d). When we used the predominant pattern to define the strata, both scoring methods yielded more favorable overall survival (OS) for encapsulated metastases (Supplementary Fig. 2c, d), as expected from previous studies[4,13]. However, applying different cut-offs for encapsulation, we observed a trifurcation of the strata with improving outcomes from <33% over 33–<100% to 100% encapsulation (Supplementary Fig. 2e, f); this trifurcation was most distinct in patients that had received neoadjuvant chemotherapy and when using extended scoring (Supplementary Fig. 2e, f). These data showed that systematized measurements affect the granularity of growth pattern scoring, which alters the classification of some cases. More fine-grained scores might improve prognostic stratification and the resulting trifurcation in the survival data challenged the concept that only those patients with fully encapsulated tumors have favorable survival.

The differences between standard and extended scoring prompted us to complete digital annotations for the entire cohort comprising n = 263 CRLM patients, for which a total of n = 897 WSIs were evaluated, resulting in n = 60,878 individual annotations. For n = 231 patients, a predominant growth pattern could be determined, after excluding patients with complete regression whose growth patterns were not assessed according to the guidelines[2], and those patients with no available representative sections. In the final series, n = 122 (53%) patients had metastases with predominantly desmoplastic growth, n = 105 (45%) had predominantly replacement growth, and n = 4 (2%) patients had predominantly "pushing" metastases (a rare pattern where perimetastatic hepatocytes appear flattened), similar to previously reported distributions[2,13]. No difference in OS between male and female patients was observed (log-rank p = 0.99), and, therefore, sex was not considered in further analyses. Over the cumulative invasion front of all tumors, the encapsulated pattern was most frequent (55%), followed by replacement (43%) and pushing (2%). As expected, predominantly encapsulated CRLM had a significantly better OS and liver-specific relapse-free survival (hepatic-RFS, hRFS), compared to predominantly replacement metastases and predominantly pushing metastases (Fig. 2a).

Our annotations allowed high-resolution stratification of growth pattern fractions. Using the previously suggested cut-offs[2] on the digital annotations revealed the proportion-dependency of the outcome on growth pattern fractions, such that partly encapsulated metastases formed a group with intermediate OS (Supplementary Fig. 3a). Survival curves best approximated a trifurcation into low (0–33%), medium (33–<100%) and high (100%) proportions of encapsulation (Fig. 2b). Accordingly, increasing fractions of tumor encapsulation were associated with a lower risk for death (Cox-proportional hazard model for encapsulation in 0.1 fraction increments: hazard

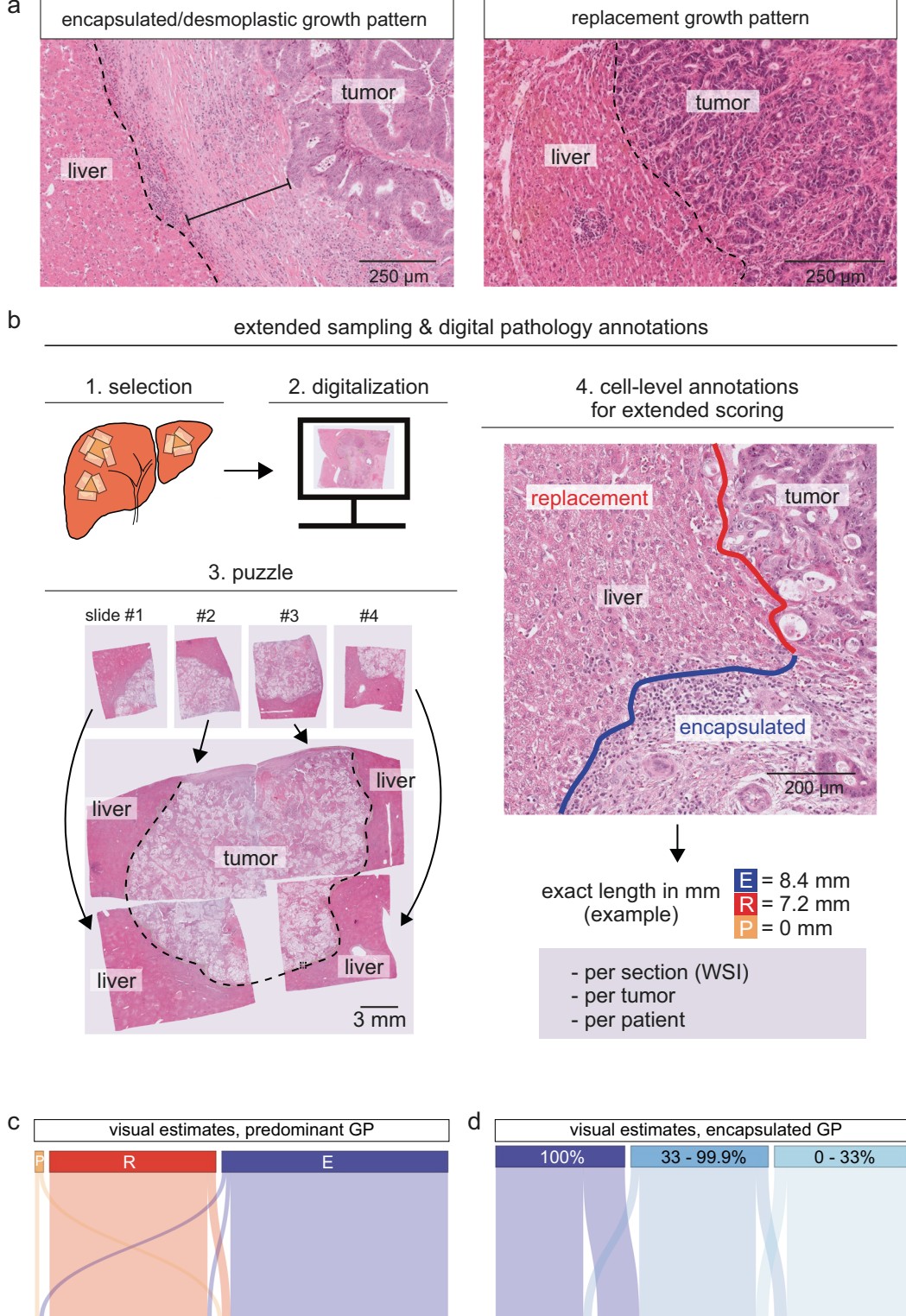

**a** encapsulated/desmoplastic growth pattern | replacement growth pattern

**b** extended sampling & digital pathology annotations

1. selection  2. digitalization  4. cell-level annotations for extended scoring

3. puzzle

exact length in mm (example)
E = 8.4 mm
R = 7.2 mm
P = 0 mm

- per section (WSI)
- per tumor
- per patient

**c** visual estimates, predominant GP / extended scoring, predominant GP

**d** visual estimates, encapsulated GP / extended scoring, encapsulated GP

ratio [HR] for death = 0.91, 95% confidence interval [CI] 0.87–0.95, $p = 2e{-}04$, Wald-test) and relapse (HR for hRFS = 0.92, 95%-CI: 0.88–0.96, $p = 4e{-}05$, Wald-test). As expected due to the mutual exclusivity of encapsulated and replacement regions, stratification into three strata of replacement growth (0%, >0–<66%, 66–100%) yielded comparable results (Supplementary Fig. 3b). Uni- and multivariate analyses of the three strata for encapsulation confirmed the lower risk for death and liver relapse in both medium (hazard ratios [HR] = 0.57 and 0.67 respectively in univariate analyses) and high (HR = 0.38 and 0.44 respectively in univariate analysis) proportion strata (Fig. 2c, d).

Hence, the degree of encapsulation, rather than the mere presence of replacement growth, has prognostic value for CRLM patients after surgery.

**Fig. 1 | Extended versus standard scoring of colorectal cancer liver metastases.**
**a** "Desmoplastic" or "encapsulated" pattern (left panel): A stromal rim separates tumor cells and liver parenchyma. The dashed line indicates the liver-capsule border, the continuous line indicates the distance from the liver to the nearest tumor across the capsule. Replacement pattern (right panel): Cancer cells are in contact with hepatocytes, appearing to replace them. The dashed line indicates the liver-tumor border. Images are representative of all the metastases in the cohort (*n* > 500). **b** Schematic of slide selection and extended annotations on whole slide images (WSIs). 1 (Selection): All available sections from each patient are reviewed using light microscopy of all available hematoxylin & eosin stains, and of all immunohistochemical stains, if available. Macroscopic pictures are used as guidance (related to Supplementary Fig. 1). The sections representing the largest metastasis diameter, maximizing the representation of the tumor-liver interface are selected from archived tissue, and slides without liver parenchyma as well as overlapping slides are excluded. 2 (Digitalization): Selected slides are digitized to WSIs. 3 (Puzzle): Multiple WSIs from the selected slides are combined, and final verification of correct selection (most complete representation and avoidance of overlaps) is performed; non-digitized slides are revisited if significant parts of the tumor-liver interface are missing. The dashed black line represents the tumor-liver interface. 4: (Annotation): Each slide is manually annotated for each growth pattern, indicated by the lines (red: replacement, blue: encapsulated). **c** Sankey plot depicting reclassifications resulting from visual estimates compared to extended WSI-based scoring. **d** Sankey plot showing reclassifications based on the indicated percentages of desmoplastic encapsulation; *n* = 94 patients for (**c**) and (**d**); source data are provided with the Source Data file.

## The perimetastatic capsule bears similarity to benign liver fibrosis

The proportion-dependency of survival on encapsulation supported a probabilistic model, where the balance of replacement growth and encapsulation determines tumor aggressiveness. However, the link between decreased tumor aggressiveness and the presence of the encapsulating rim remains elusive. We hypothesized that a deeper understanding of the rim's cellular composition could help elucidate its origin. To phenotype the stromal cells in the rim, we analyzed the expression of stromal protein markers from the liver parenchyma to the central tumor stroma. We quantified stains of the liver fibrosis markers, low-affinity nerve growth factor receptor (NGFR)[19,20] and alpha-smooth muscle actin (ASMA)[21], which revealed the zonation of the capsule, such that expression of both NGFR and ASMA increased from the metastasis edge towards the rim-liver interface (Fig. 3a–c). NGFR expression was highest in the outer rim, resembling the profile of benign fibrosis[19,20]. This prompted us to revisit stromal markers in livers with benign fibrotic conditions. We found that the fibrous stroma in focal nodular hyperplasia (Supplementary Fig. 4a) and cholangitis (Supplementary Fig. 4b) was NGFR+, akin to the outer region of the capsule. In cirrhosis, we observed a similar NGFR+ stroma, condensing with higher degrees of liver plate atrophy (Supplementary Fig. 4c, d).

To thoroughly characterize the stromal cells in the capsule, we employed multiplex immunofluorescence (m-IF) for multiple stromal markers (Fig. 3d and Supplementary Fig. 5a). Quantitation along the trajectory from the tumor to the perimetastatic liver confirmed an ASMA protein gradient from the inner to the outer rim (Fig. 3e, f). In parallel, the expression of platelet-derived growth factor receptor alpha (PDGFRa), a marker for scar-associated mesenchymal cells that expand in fibrosis[22], increased towards the outer rim. In contrast, fibroblast-activation protein (FAP), which marks injury-induced fibroblasts that dominate hepatic collagen production in later rather than early fibrosis stages[23], peaked in the inner part of the rim and dominated the intrametastatic stroma (Fig. 3e, f). Expression levels of the matrix-remodeling enzyme, lysyl oxidase-like 2 (LOXL2)[24], phosphorylated signal transducer and activator of transcription factor 3 (p-STAT3), which marks subsets of immunosuppressive fibroblasts in cancer[25], and of tissue factor (TF), which marks subendothelial tissues[26], were low in the capsule (Fig. 3e). The stroma surrounding portal triads, which constitute a major hub for benign liver fibrotic conditions, had a marker profile similar to that of the outer rim, as it showed high expression of NGFR (Fig. 3c). Multiplex-IF revealed further similarities dominated by high expression of ASMA, and by medium-to-high expression of PDGFRa, as well as low expression of FAP and PDGFRb (Supplementary Fig. 5b). Multiplex RNA in situ hybridization (ISH) for the stromal markers, *COL1A1* (collagen 1), *DCN* (decorin), *FN1* (fibronectin-1), *PDGFRA*, SPP1 (osteopontin), and *THY* (Thy-1 cell surface antigen) confirmed zonal expression patterns of the capsule, such that its outer part was enriched for *COL1A1*, *DCN*, *FN1*, and *PDGFRA* (Supplementary Fig. 5c, d).

These results unveiled the zonation of the capsule, dominated by increasing gradients of NGFR and ASMA from the tumor to the liver and resembling scar-associated portal-like fibrosis at the capsule-liver interface.

## Hepatic injury surrounds the perimetastatic capsule

Spatial transcriptomics of benign fibrosis has revealed clusters of macrophages, ASMA+ stromal cells, and injured hepatocytes expressing acute-phase proteins, such as C-reactive protein (CRP)[27]. Such fibroinflammatory foci are often accompanied by so-called ductular reactions, which represent nests of newly formed bile duct-like aggregates[28,29] and by increased hepatocyte expression of cytokeratin 18 (CK18)[30]. On the vascular level, liver injury can induce capillarization of the liver sinusoids, for which cluster of differentiation (CD) 34 is an established marker[31].

To chart these injury-related cellular constituents in and around the capsule, we first generated protein expression maps of the CRP and CK18 across the capsule and the peritumoral liver. We observed a gradient of their expression starting at the rim-liver interface (Fig. 4a–c) and declining within the liver parenchyma, suggesting a localized perimetastatic hepatocyte injury. Staining for cellular components of hepatic fibroinflammation with m-IHC, we found macrophage accumulation (CD68 expression), ductular reactions/bile duct remnants[32] (CK7 expression, Fig. 4d–f), and CD34+ cells indicating vascular remodeling predominantly at the outer edge of the rim, consistent with a hepatic injury reaction that was strongest at the rim-liver interface.

## Presence of liver parenchymal remnants in the capsule

Our results identified the outer capsule as a hotspot of liver injury, reminiscent of benign-like fibrosis, which suggested that it originates from the resident parenchymal hepatic stroma. However, on routine pathological evaluation of the capsule, we only rarely observed hepatocytes. To investigate their presence, we used mRNA in situ hybridization (ISH) for the hepatocyte marker albumin (*ALB*). ISH revealed *ALB*-expressing cells in the perimetastatic capsule; however, these cells were smaller than normal hepatocytes and had lost their typical cuboidal morphology, likely reflecting regenerative changes toward a hepatic progenitor-like phenotype (Fig. 5a)[28].

Liver lobules, the functional units of the liver parenchyma, contain portal triads at their corners, including the portal vein (V), the hepatic artery (A), and the bile duct (BD). Portal triads are nodes for fibrosis development in hepatitis and primary biliary diseases, as well as after parenchymal scarring following hepatocyte atrophy[33] and necrosis[34]. Portal triads are embedded in a specific, ASMA^{high};NGFR^{high};CD34^{high} stroma[2]. In clinical routine, we observed different degrees of portal triad atrophy: Aside from complete triads (A + BD + V), remnants with the artery and bile duct (A + BD) can be seen, suggesting an obliterated vein[35], as well as those with only the artery (A) in its characteristic stroma (Supplementary Fig. 6a)[2]; these structures can be

## Table 1 | Clinical characteristics of all patients in the cohort

| Characteristic | n = 263 | 95% CI |
|---|---|---|
| **Sex** | | |
| Female | 100 (38%) | 32%, 44% |
| Male | 163 (62%) | 56%, 68% |
| **Age (years)** | 67 (59, 73) | 64, 67 |
| **Neoadjuvant chemotherapy** | 165 (63%) | 57%, 69% |
| Unknown | 2 | |
| **ASA classification** | | |
| 1 | 98 (37%) | 31%, 43% |
| 2 | 52 (20%) | 15%, 25% |
| 3 | 113 (43%) | 37%, 49% |
| **WHO performance status** | | |
| 0 | 227 (87%) | 82%, 90% |
| 1 | 33 (13%) | 8.9%, 17% |
| 2 | 2 (0.8%) | 0.13%, 3.0% |
| Unknown | 1 | |
| **Charlson comorbidity index** | | |
| 6–8 | 135 (52%) | 45%, 58% |
| 9 | 75 (29%) | 23%, 35% |
| 10–14 | 52 (20%) | 15%, 25% |
| Unknown | 1 | |
| **Location of primary tumor** | | |
| Left | 201 (77%) | 71%, 82% |
| Right | 60 (23%) | 18%, 29% |
| Unknown | 2 | |
| **Tumor stage (T)** | | |
| 0 | 3 (1.3%) | 0.33%, 4.0% |
| 1 | 2 (0.8%) | 0.15%, 3.4% |
| 2 | 31 (13%) | 9.2%, 18% |
| 3 | 143 (61%) | 54%, 67% |
| 4 | 57 (24%) | 19%, 30% |
| Unknown | 27 | |
| **Nodal stage (N)** | | |
| 0 | 84 (36%) | 30%, 42% |
| 1 | 93 (40%) | 33%, 46% |
| 2 | 57 (24%) | 19%, 30% |
| Unknown | 29 | |
| **Resected primary tumor** | 247 (95%) | 91%, 97% |
| Unknown | 2 | |
| **Radicality, primary tumor** | | |
| R0 | 201 (97%) | 93%, 99% |
| R1 | 6 (2.9%) | 1.2%, 6.5% |
| R2 | 1 (0.5%) | 0.03%, 3.1% |
| Unknown | 55 | |
| **Meta-/ synchronous metastasis** | | |
| Metachronous | 128 (51%) | 44%, 57% |
| Synchronous | 124 (49%) | 43%, 56% |
| Unknown | 11 | |
| **Number of liver metastases** | | |
| <4 | 202 (82%) | 77%, 87% |
| 4+ | 43 (18%) | 13%, 23% |
| Unknown | 18 | |
| **Mean number of liver metastases** | 2 (1, 3) | 2.1, 2.8 |
| Unknown | 18 | |

## Table 1 (continued) | Clinical characteristics of all patients in the cohort

| Characteristic | n = 263 | 95% CI |
|---|---|---|
| **Max. diameter of individual metastasis** | | |
| <5 cm | 217 (83%) | 77%, 87% |
| 5+ cm | 46 (17%) | 13%, 23% |
| **Resection margin (liver metastasis)** | | |
| <1 mm | 96 (41%) | 35%, 48% |
| 1 mm + | 137 (59%) | 52%, 65% |
| Unknown | 30 | |
| **Sum of metastasis diameters (cm)** | 3.0 (1.7, 6.2) | 4.1, 5.2 |
| **Metastases regression [%]** | 36 (18, 55) | 34, 40 |
| Unknown | 29 | |
| **Cases with KRAS mutation** | 42 (41%) | 32%, 51% |
| Unknown | 161 | |
| **MSI-high cases** | 1 (7.1%) | 0.37%, 36% |
| Unknown | 249 | |
| **Cases with BRAF mutation** | 1 (1.2%) | 0.06%, 7.6% |
| Unknown | 182 | |

Column 1: number of patients or median with interquartile range in brackets.
Column 2, *CI* confidence interval, *ASA* American Society of Anesthesiologists, *WHO* World Health Organization, *KRAS* Kirsten rat sarcoma virus gene, *MSI* microsatellite instability, *BRAF* B-Raf Proto-Oncogene.

approximated based on hematoxylin & eosin stains by their characteristic morphology.

To test the hypothesis that the perimetastatic stroma is a reparative remnant of injured liver plates, we revisited those cases in our cohort with >95% encapsulation (n = 49 cases, corresponding to n = 141 WSIs) and quantified portal triad remnants (Fig. 5b). The results revealed remnants in the fibrotic rim in all cases at varying frequencies (Supplementary Fig. 6b). We found similar degrees of portal atrophy in primary liver tumors, suggesting that the continuum of atrophy is not specific to metastases (Supplementary Fig. 6c). When we analyzed the spatial distribution of A and A + BD remnants separately, we found that A remnants were more frequent towards the tumoral side, while A + BD remnants were enriched at the liver side of the capsule (Fig. 5c, d). We also observed portal triad remnants in the center of encapsulated metastases, which occurred at a lower density than in the rim (Supplementary Fig. 6d, e) and which suggested previous replacement-type growth, rather than continuous encapsulated expansion.

Visual scoring of the growth patterns in series of non-colorectal liver metastases has consistently demonstrated a survival benefit for patients with encapsulated metastases[5,6,9,36], hinting at a unifying mechanism of capsule formation. To investigate liver parenchyma remnants in encapsulated metastases from other primary tumors and in primary liver cancers, we analyzed hematoxylin & eosin-stained slides and stromal marker stains of metastases from non-colorectal primary tumors and from primary liver cancers. In metastases of adenocarcinomas of the pancreas and gallbladder, as well as in intrahepatic cholangiocarcinoma, we found remnants of portal tracts in the capsule on hematoxylin & eosin-stained sections (Supplementary Fig. 7a–c). In breast cancer and melanoma liver metastases, and in hepatocellular carcinoma, where stromal stains were available, we additionally observed an NGFR stromal gradient across the capsule, resembling that seen in CRLM (Supplementary Fig. 7d–f). Multiplex-IF of intrahepatic cholangiocarcinoma (Supplementary Fig. 7g) revealed stromal protein gradients similar to

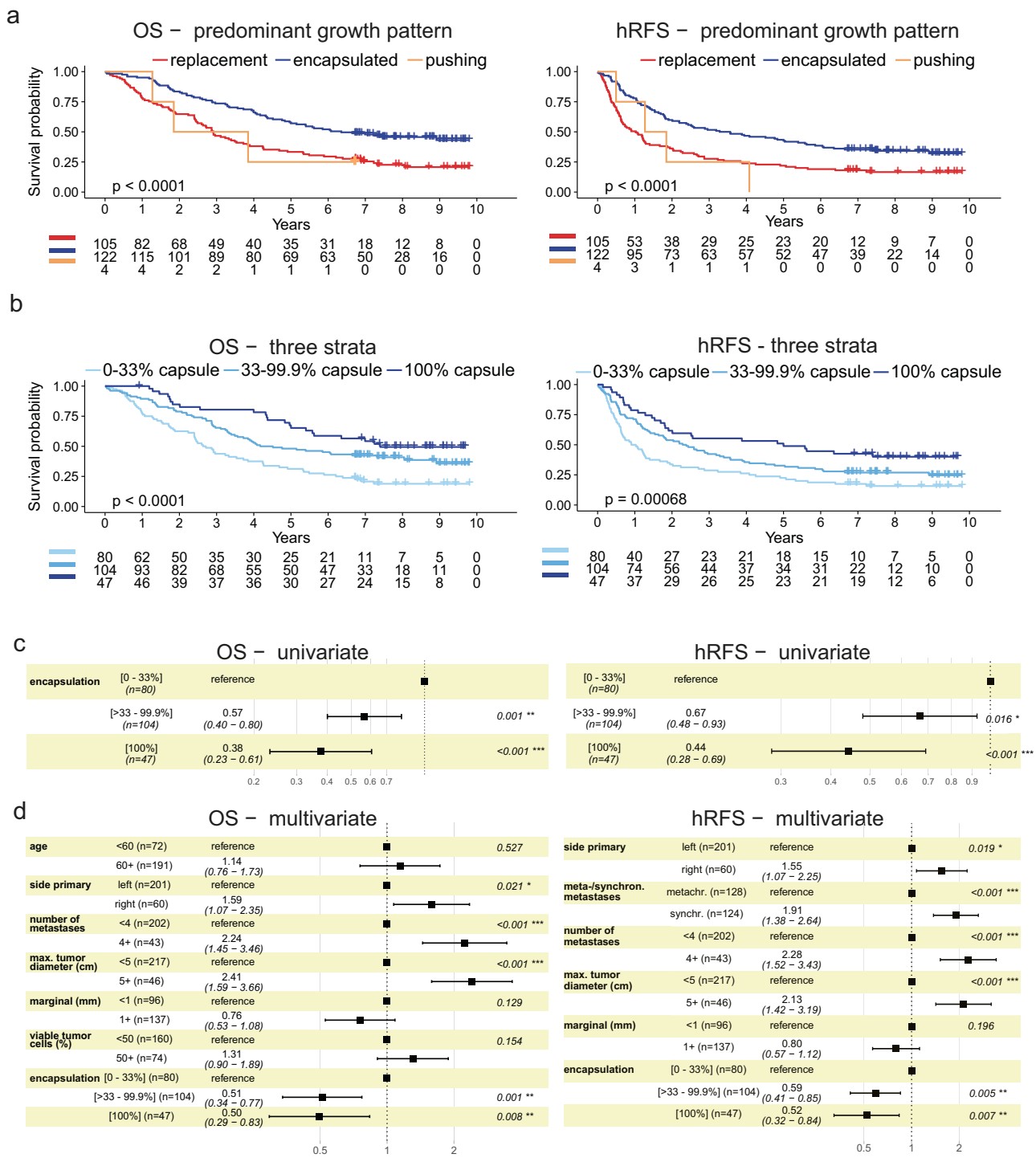

**Fig. 2 | Prognostic trifurcation by fraction of encapsulation. a** Overall survival (OS, left) and liver-specific (hepatic) relapse-free survival (hRFS, right), when scored by the predominant growth pattern, replacement, encapsulated, or pushing. **b** OS and hRFS stratified by the indicated proportions of encapsulation. Two-sided log-rank *p*-values given in the plots, (**a**) and (**b**). OS – predominant growth pattern, *p* = 0.000075; hRFS – predominant growth pattern, *p* = 0.000097; OS – three strata, *p* = 0.000038; hRFS – three strata, *p* = 0.000676. **c** Univariate Cox proportional hazard regression using three strata of encapsulation, OS left; hRFS right. OS – 100% encapsulation, *p* = 0.000063; hRFS – 100% encapsulation, *p* = 0.000312. **d** Multivariate Cox proportional hazard analysis using encapsulation in three strata; OS left; hRFS right. Hazard ratio (HR) with 95% confidence interval shown in brackets. Two-sided Wald-test *p*-value, asterisks represent significance levels; ***p* < 0.001, **p* < 0.01, *p* < 0.05. OS – number of metastases, *p* = 0.000298; OS – max. tumor diameter (cm), *p* = 0.000037; hRFS – meta-/synchronous metastases, *p* = 0.000107; hRFS – number of metastases, *p* = 0.000070; hRFS – max. tumor diameter (cm), *p* = 0.000241.

those observed in CRLM, which were characterized by ASMA, NGFR, and PDGFRa at the outer edge, and FAP and PDGFRb at the inner side of the rim (Fig. 5e).

Together, these data identified the anatomical constituents of liver-derived stroma in the capsule, which followed a gradient towards more advanced liver atrophy on the tumoral side.

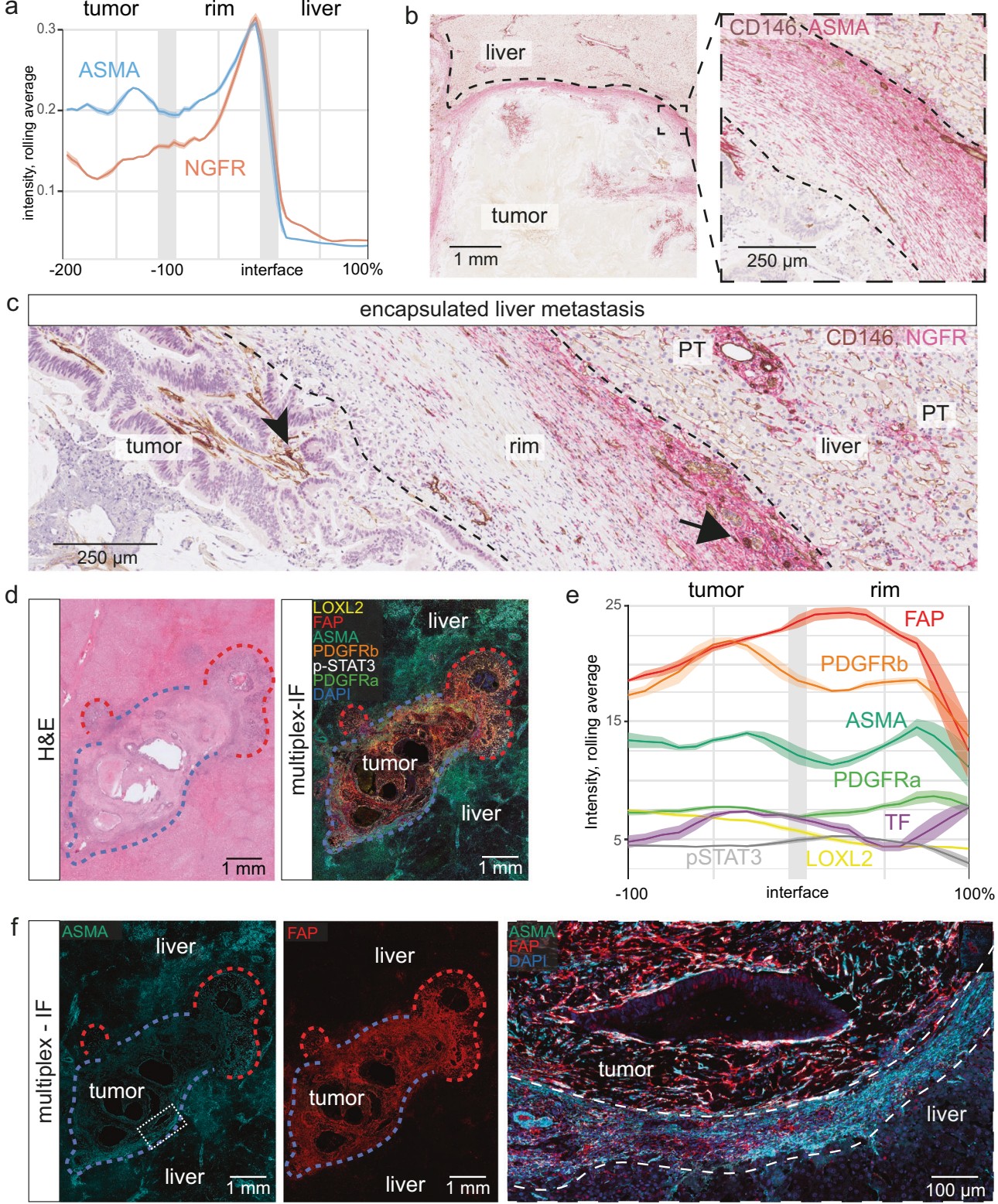

### Increased encapsulation after neoadjuvant chemotherapy

A recent study has reported an increased frequency of encapsulated metastases after neoadjuvant chemotherapy[15], potentially linking chemotherapy-induced tumor cell death and encapsulation. However, previous studies had not reported similar findings[13]. Using the extended scoring of our cohort, we found that the frequency of encapsulation was significantly higher in patients who had received neoadjuvant chemotherapy than in those who had not (Fig. 6a). Our

annotation data further allowed us to quantify the degree of encapsulation on each slide of each metastasis for every patient (Supplementary Fig. 8a), which illustrated the increase in encapsulation after chemotherapy (Fig. 6b).

Within the patient group that had received neoadjuvant treatment, there was a strong negative correlation between the fraction of viable tumor cells and encapsulation (Spearman's $r = -0.52$, $p = 3.5e{-}11$, Fig. 6c). This association was present but weaker in the chemonaive

**Fig. 3 | Similarity of the capsule to benign fibrosis. a** Quantification of nerve growth factor receptor (NGFR) and alpha-smooth muscle actin (ASMA) protein expression in tumor, perimetastatic capsule, and liver, based on immunohistochemistry stains; the lines represent rolling averages, the areas represent the 95% confidence interval (CI; *n* = 6 patients). Note increased expression of NGFR and ASMA towards the outer, perihepatic edge of the rim. **b** Representative example (of *n* = 6 metastases from *n* = 6 patients) of ASMA staining (red), with cluster of differentiation (CD)146 (brown) marking, among other structures, ductular reactions. **c** Representative example (of *n* = 6 metastases from *n* = 6 patients) of NGFR⁺ gradient; note the strong NGFR expression (red) increasing towards the liver. CD146 (brown); arrowhead: intratumoral stroma; arrow: portal tract remnant in the capsule. PT: portal triad; note the NGFR⁺ portal stroma. **d** Representative hematoxylin and eosin stain (left panel) and corresponding image of 6-plex immunofluorescence of the indicated proteins, using 4′,6-diamidino-2-phenylindole (DAPI) for nuclear staining (right panel). Representative of *n* = 6 metastases from *n* = 6 patients; different patients from (**a**)–(**c**). **e** Quantification of immunofluorescence

protein expression in the indicated regions; the lines represent rolling averages, the areas represent 95% CI (*n* = 6, for which at least *n* = 4 cases were quantified per stain; different encapsulated cases than shown in [**a**]). Tissue factor (TF) was analyzed in a separate run and is not included in the representative image (**d**). **f** Multiplex immunofluorescence of the sample shown in (**d**) for ASMA (left), fibroblast-associated protein (FAP, right), and for a composite ASMA, FAP, DAPI (large panel to the right); FAP dominates in the tumor center, ASMA increases towards the liver side of the capsule. Growth pattern indicated in red (replacement) or blue (encapsulated). The dashed white line marks the inner and outer borders of the capsule. Source data for the plots in (**a**) and (**e**) are provided with the Source Data file. Thresholding for low-intensity values was used in the fluorescent images in panels (**d**) and (**f**) to reduce autofluorescensce background in the channels depicting ASMA (480 nm) and PDGFRa (520 nm). Quantification for these channels was done by subtracting the signal from the automatically generated autofluorescence channel, without thresholding.

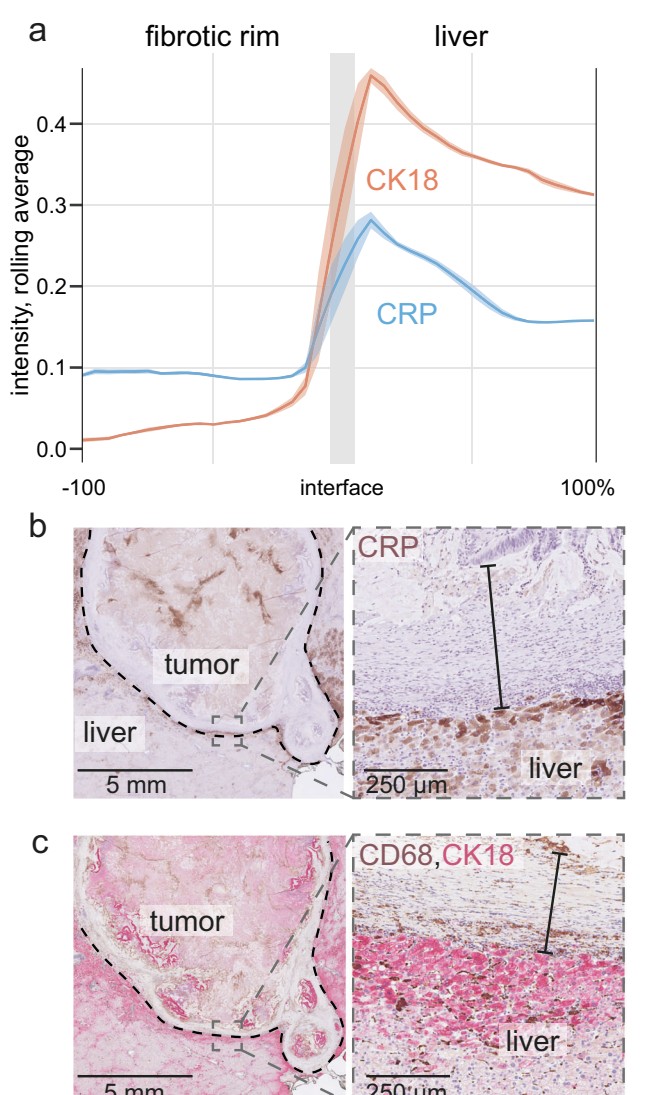

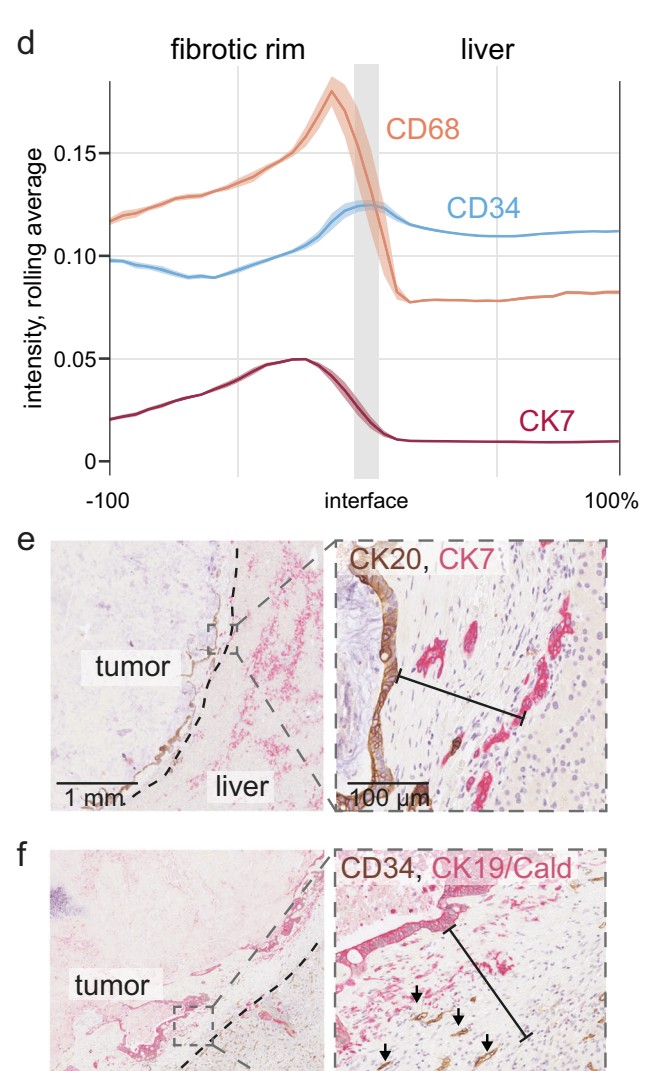

**Fig. 4 | Protein expression gradients in the capsule. a** Immunohistochemistry (IHC)-based quantification of C-reactive protein (CRP) and cytokeratin 18 (CK18) across the entire perimeter of the fibrotic rim and in the perimetastatic liver parenchyma. Average values from *n* = 6 metastases, for which the entire desmoplastic rim was measured; the lines represent rolling averages, the areas represent 95% confidence intervals (CIs). **b, c** Representative examples of CRP, CK18/cluster of differentiation (CD) 68 IHC staining, as indicated. The dashed line marks the rim's outer border; the solid line denotes the rim's thickness. **d** Quantification of CD34

(endothelium), CD68 (macrophages), and CK7 (bile ducts and ductular reactions); rolling averages/CIs of results from *n* = 6 patients are shown. **e–f** Representative images of the indicated IHC stains. A dashed line marks the outer border of the rim; the solid line denotes the thickness of the rim, and arrows point at CD34⁺ vessel remnants in the rim. Panels (**b**), (**c**), (**e**), (**f**) are representative of *n* = 6 metastases from *n* = 6 patients, corresponding to the quantifications shown in panels (**a**) and (**d**). Source data for the plots in (**a**) and (**d**) are provided with the Source Data file.

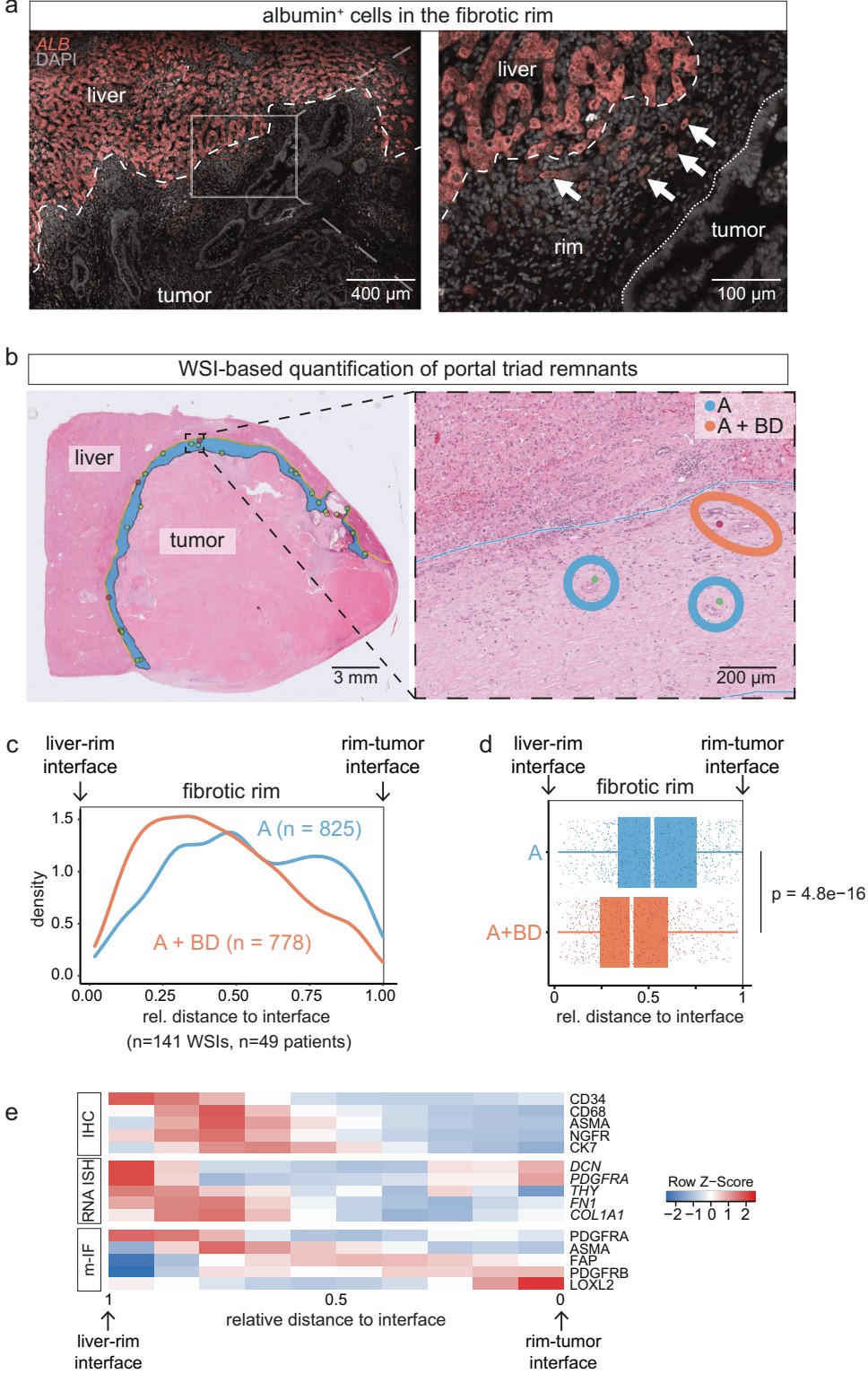

**Fig. 5 | Liver parenchymal remnants in the capsule. a** mRNA in situ hybridization (ISH) of an encapsulated CRLM for albumin (*ALB*), nuclear stain with 4′,6-diamidino-2-phenylindole (DAPI). Arrows: *ALB*⁺ cells in the rim. Dashed line: liver-rim interface; dotted line: metastasis border towards the liver. Representative image of *n* = 4 encapsulated metastases. **b** Left: Whole-slide image (hematoxylin & eosin), annotated for the capsule and portal tract (PT) remnants: blue area marks the rim; red dots mark PTs consisting of artery and bile duct (A + BD), green dots mark PTs consisting of artery remnants (A). Right: Magnification, blue line marks the rim-liver interface, A remnants highlighted in blue, A + BD remnant highlighted in orange; representative of *n* = 49 patients. **c** PT density in the desmoplastic rim of encapsulated metastases from *n* = 49 patients, based on *n* = 141 whole-slide images

(WSIs). **d** Box-and-Whisker plot of the distributions of PT arteries (A) and PT remnants consisting of artery and bile duct (A + BD). Result from two-sided Wilcoxon rank sum test is presented in the panel; based on the same data as (**c**). Median (line), interquartile range (box), minimum and maximum values within 1.5 times the IQR from the first and third quartiles (whiskers) and individual datapoints are shown. **e** Heatmap summarizing zonated expression patterns derived from different analysis modalities: immunohistochemistry (IHC, corresponding to Figs. 3a, 4d), multiplex RNA ISH (corresponding to Supplementary Fig. 5d) and multiplex immunofluorescence (m-IF, corresponding to Fig. 3e). Source data for the plots in (**c**, **d**) are provided with the Source Data file.

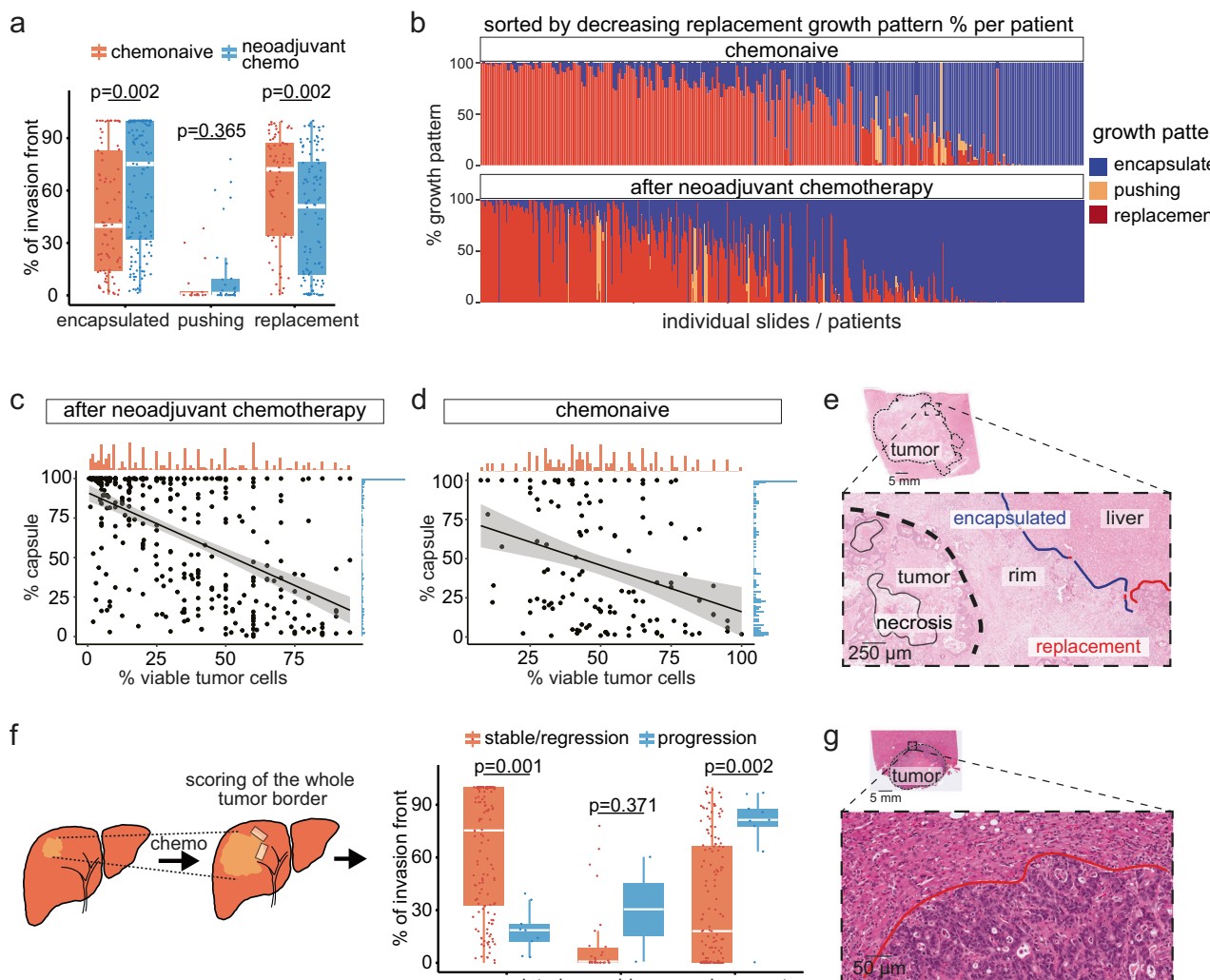

**Fig. 6 | Impact of neoadjuvant chemotherapy on growth pattern frequency.**
**a** Change in percentages of encapsulated, pushing, and replacement patterns in patients who received chemotherapy *vs.* those who did not; Box-and-Whisker plot with median (line), interquartile range (box), minimum and maximum values within 1.5 times the IQR from the first and third quartiles (whiskers) and individual datapoints are shown. Result from two-sided Wilcoxon-test shown in the panel.
**b** Distribution of the growth patterns per slide in chemonaive patients and patients who received neoadjuvant chemotherapy. Note the difference in encapsulation (blue). **c, d** Correlation between the percentages of encapsulated growth pattern and percentages of viable tumor cells in chemotherapy-treated (**c**) vs. untreated (**d**) patients. Individual datapoints and linear regression model (black line) with confidence intervals (in gray) are shown in (**c**) and (**d**). **e** Example of a treated metastasis

(hematoxylin & eosin; representative of metastases from *n* = 165 patients), illustrating viable tumor cells remote from the surrounding liver parenchyma. The black circled structures denote necrotic tumor, the blue line the encapsulated, and the red line the replacement growth pattern. **f** Growth pattern frequencies of patients with stable disease or regression (orange) vs patients that progressed on neoadjuvant chemotherapy (blue); Box-and-Whisker plot, median (line), interquartile range (box), minimum and maximum values within 1.5 times the IQR from the first and third quartiles (whiskers) and individual datapoints are shown. Result from two-sided Wilcoxon-test shown in the panel. **g** Representative image (of *n* = 8 patients; hematoxylin & eosin) of a treated metastasis that progressed during chemotherapy, red line indicates (replacement-type) invasion front. Source data for the plots in (**a**), (**c**), (**d**), and (**f**) are provided with the Source Data file.

group (Spearman's *r* = −0.24, *p* = 0.028, Fig. 6d). Histology confirmed that the remaining viable tumor cells were located both in regions of fibrotic encapsulation and in those of replacement-type growth (Fig. 6e).

At our hospital, surgical resection is generally not performed when radiological tumor progression is noted during neoadjuvant chemotherapy. Hence, metastases that progress upon chemotherapy are largely absent from the neoadjuvant group, potentially skewing the data towards the encapsulated type in treated patients. However, resection is occasionally performed despite progression, for example in young patients or in case of localized disease. We hypothesized that tumors with a high capability to invade the liver, as demonstrated by progression despite chemotherapy, would be dominated by replacement growth. Hence, we identified those patients that underwent

resection despite tumor progression on chemotherapy in an extended cohort (from 2012 to 2020). Out of *n* = 656 patients, *n* = 10 (1.5%) were operated on despite progression, and we could retrieve material from *n* = 9 patients for digital scoring, which revealed predominantly non-encapsulated growth in all of them (Supplementary Fig. 8b). The replacement pattern was significantly more frequent in patients that had progressed on chemotherapy compared to patients who had stable disease or regression as assessed radiologically, while encapsulation was less frequent (Fig. 6f, g).

Next, we used these data to approximate the impact of the potential selection bias that could stem from abstaining from resection in case of tumor progression. We identified all patients that had started preoperative chemotherapy, but whose metastases were not resected because of progression. Data for this metric were available for

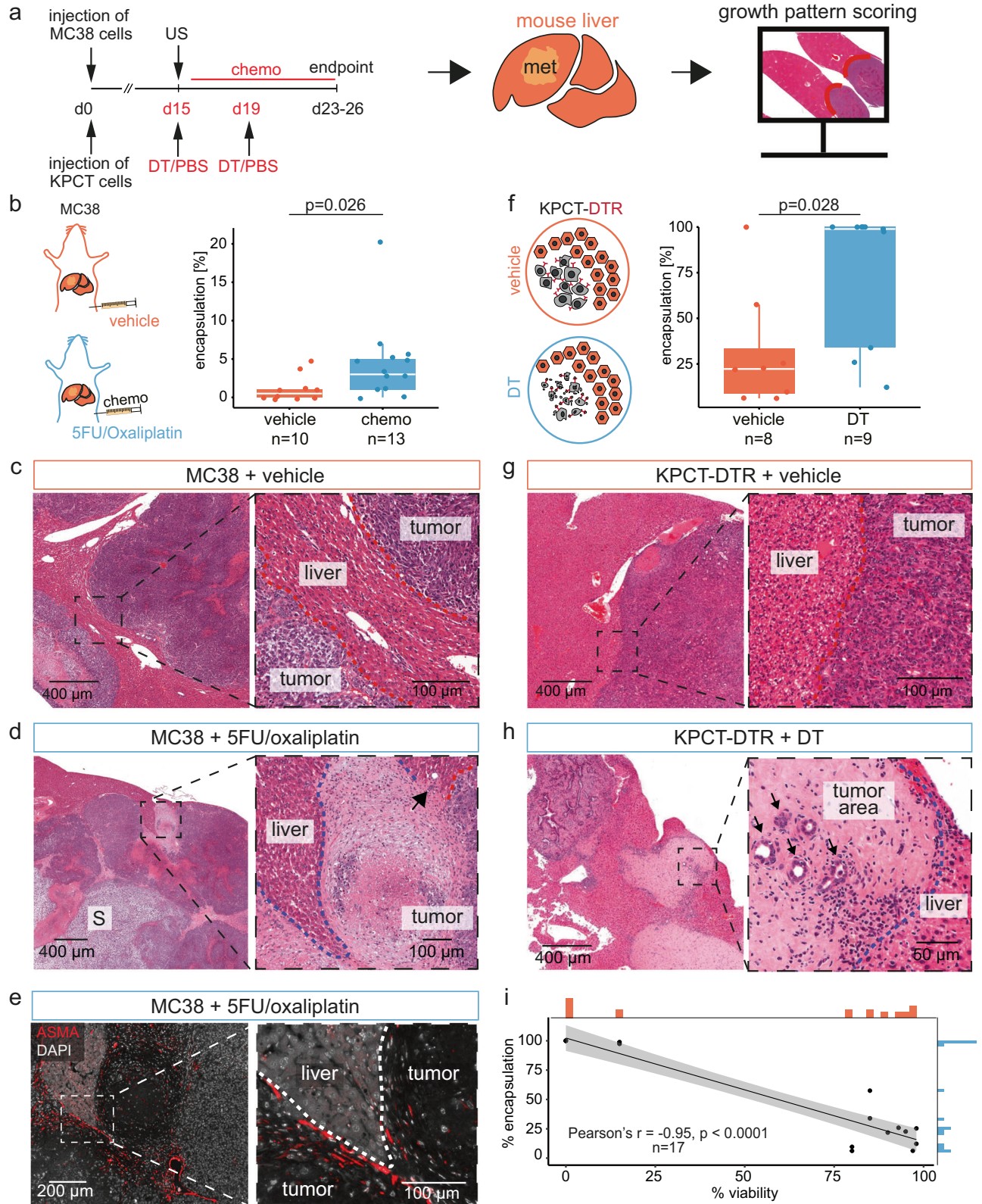

the years 2013–2018; out of $n = 409$ patients, the operation was canceled because of progression in $n = 32$ patients (7.8%), for whom, consequently, no histological material was available. We then repeated the comparison of the growth pattern frequencies in chemonaive vs. treated patients by adding to the treated group a corresponding fraction of 7.8% of patients based on the growth pattern measurements of the tumors that had progressed during chemotherapy (i.e.,

dominated by replacement growth). We found that despite the addition of these hypothesized progressors, the difference in favor of the desmoplastic pattern remained significant (Supplementary Fig. 8c). When we compared the clinicopathological characteristics of patients receiving neoadjuvant treatment to those who did not, we found that patients who had received chemotherapy were younger and had significantly more metastases than chemonaive patients. However, the

**Fig. 7 | Induction of perimetastatic encapsulation by chemotherapy and tumor cell ablation in mice. a** Experiment schematic. US: ultrasound. **b** Percentages of metastasis encapsulation in mice injected with MC38 cells and treated with either vehicle (saline) or 5-fluorouracil and oxaliplatin (5FU/Oxa); $n = 30$ mice were injected with MC38 cells, of which $n = 24$ mice had visible tumors on ultrasound (day 15); those were included and randomized into chemotherapy ($n = 14$) or saline ($n = 10$) treatment groups. No tumor was seen on histology of one mouse in the treatment group. Box-and-Whisker plot, result from two-sided Wilcoxon-test shown in the panel, median (line), interquartile range (box), minimum and maximum values within 1.5 times the IQR from the first and third quartiles (whiskers). **c** Histomorphology (hematoxylin & eosin) of mice bearing MC38 metastases treated with vehicle (saline) and **d** treated with 5FU/Oxa. Red dashed line indicates replacement, blue indicates encapsulated pattern. S: sarcomatoid growth. **e** Representative immunofluorescence for alpha-smooth muscle actin (ASMA, red) in an encapsulated, treated metastasis. The dashed lines indicate the rim-liver

border. Representative of stains on metastases from $n = 3$ 5FU/Oxa treated mice. **f** Percentages of metastasis encapsulation in mice treated with vehicle (phosphate-buffered saline, PBS) or diphtheria toxin (DT). $n = 17$ KPCT-DTR tumor-bearing mice (of $n = 30$ initially injected with KPCT cells) were treated either with PBS ($n = 8$) or DT ($n = 9$). Box-and-Whisker plots, result from two-sided Wilcoxon-test shown in the panel, median (line), interquartile range (box), minimum and maximum values within 1.5 times the IQR from the first and third quartiles (whiskers). **g** Representative histomorphology (hematoxylin & eosin) of liver metastases from mice treated with vehicle or **h** DT. Dashed lines as in (**c**, **d**). **i** Correlation of annotated percentages of encapsulated growth pattern and percentages of viable tumor cells in all mice $n = 17$ of this experiment, irrespective of treatment group. Individual datapoints and linear regression model (black line) with confidence intervals (in gray) are shown. Pearson correlation coefficient $r$ with $p$-value shown in the panel, $p = 0.0000000046$. Source data for the plots in (**b**), (**f**) and (**i**) are provided with the Source Data file.

combined sum of tumor diameters was similar, indicating comparable tumor burden in both groups (Supplementary Table 1).

These results were in line with the interpretation that impaired tumor fitness, which can be assumed in responders to chemotherapy, facilitates the development of the capsule. Next, we asked whether an underlying pro-fibrotic state of the liver could promote capsule formation. We first analyzed clinical scorings of non-tumorous liver pathology (fibrosis, inflammatory activity, presence of Periodic acid–Schiff–diastase [PAS-D] globules, and parenchymal iron deposits) in the context of the growth patterns, including all patients with either >85% encapsulated tumors ($n = 77$) or >85% replacement growth ($n = 43$ patients). Although there was a trend for increased inflammatory activity in the liver parenchyma of patients with predominantly encapsulated metastases, the differences were not significant (Supplementary Table 2). Next, we reviewed the charts of all $n = 656$ patients that underwent surgical resection for CRLM 2012–2020 and that had pre-existing, clinically established fibrotic liver disease. We identified $n = 3$ patients (0.46%), one with primary biliary cirrhosis, one with alcohol-related cirrhosis, and one with cirrhosis due to hepatitis C. This frequency was consistent with the cirrhosis prevalence in Sweden, which is around 0.67%[37]. Two of these patients had predominantly encapsulated metastases, while one had predominantly replacement growth. We assessed NGFR expression in the replacement metastasis and found NGFR+ fibrous septa surrounding regenerative hepatocyte nodules (Supplementary Fig. 9a). Tumor cells demonstrated replacement-type growth along hepatocyte plates and aligned with the fibrous stroma where they got into contact. This phenotype of hepatic plate colonization was also seen in an intrahepatic cholangiocarcinoma in a fibrotic liver (Supplementary Fig. 9b).

Together, these data suggested that a profibrotic state of the liver does not by itself induce a perimetastatic capsule; rather, reduced tumor cell ability to colonize the liver plates, for example as a result of chemotherapy, emerged as a driver for capsule formation.

### Chemotherapy and tumor-cell ablation induces perimetastatic fibrosis in mice

To experimentally test whether diminished tumor cell fitness results in capsule formation, we used mouse models of liver metastases from colorectal and pancreatic cancer primaries in mice. We injected the murine colorectal cancer cell line, MC38, into the livers of congenic, immunocompetent C57BL/6 mice with ultrasound guidance (Fig. 7a and Supplementary Fig. 10a). We confirmed the establishment of metastases with ultrasound, and on day 15 after MC38 injection, we randomized tumor-bearing mice to either receive the chemotherapy doublet 5-fluorouracil and oxaliplatin (5FU/Oxa), or vehicle (saline). 5FU/Oxa is used as first-line therapy for CRLM in humans[38], and has shown effect against MC38 cells in vivo[39]. Chemotherapy with 5FU/Oxa led to substantial weight loss (Supplementary Fig. 10b); therefore, we ended the

experiment when the humane endpoint was reached, nine days after the start of treatment. Tumors grew rapidly and had in some individuals replaced entire liver lobes. We found that chemotherapy was associated with an increased frequency of encapsulation (Fig. 7b), although we did not observe an effect of 5FU/Oxa on tumor size at the endpoint (Supplementary Fig. 10c, d), likely related to the short treatment duration and relative resistance of the tumors. In agreement with their aggressiveness, the dominating growth pattern was replacement (Fig. 7b, c). In mice that had received 5FU/Oxa, we observed areas of tumor cell necrosis, associated with the development of a capsule reminiscent of that seen in human CRLM (Fig. 7d), and positive for ASMA at the capsule-liver interface (Fig. 7e).

MC38 cells formed metastases with heterogenous morphologies and often showed sarcomatoid differentiation (Fig. 7d), which is rare in human CRLM. In a related model, based on the injection of pancreatic ductal adenocarcinoma cells derived from *LSL-Kras*[G12D/+];*LSL-TrpS3*[R172H/+];*Pdx-1-Cre;R26-LSL-tdTomato* mice (KPCT), we observed histomorphology more similar to that of human metastases (Supplementary Fig. 10e). To test whether decreased tumor cell fitness in this relevant model induces encapsulation and to achieve a stronger tumor regression than that observed after chemotherapy, we engineered KPCT cells to express the diphtheria toxin receptor (DTR)[40], which murine cells lack, thereby making KPCT cells sensitive for diphtheria toxin (DT); enrichment after transfection resulted in ~90% DTR+ cells (Supplementary Fig. 10f). In vitro, KPCT-DTR cells responded in a dose-dependent manner to DT treatment with reduced metabolic activity (Supplementary Fig. 10g). Next, we generated metastases with KPCT-DTR cells to ablate established tumors in vivo. While body weight was not affected, the reduction in tumor size was striking in most animals (Supplementary Fig. 10h–j). DT-mediated tumor ablation was associated with significantly higher fractions of encapsulation (Fig. 7f). Histology revealed replacement-type growth in most metastases of the control group (Fig. 7g), while metastases of mice treated with DT showed partial or complete regression and signs of perilesional liver reparation, including fibrosis, ductular reactions, and the influx of inflammatory cells (Fig. 7h). There was a strong negative correlation between tumor viability and the degree of encapsulation (Pearson's $r = -0.95$, $p = 4.6e^{-9}$, Fig. 7i).

Together, these data functionally linked decreased tumor cell fitness to encapsulation.

### Discussion

In this study, we sought a deeper understanding of the origin, development, and prognostic implications of the "desmoplastic" capsule in liver metastases. Based on extensive annotations, we find a strong relationship between outcome and growth pattern proportions, such that the risk of tumor recurrence and death decreases with increasing

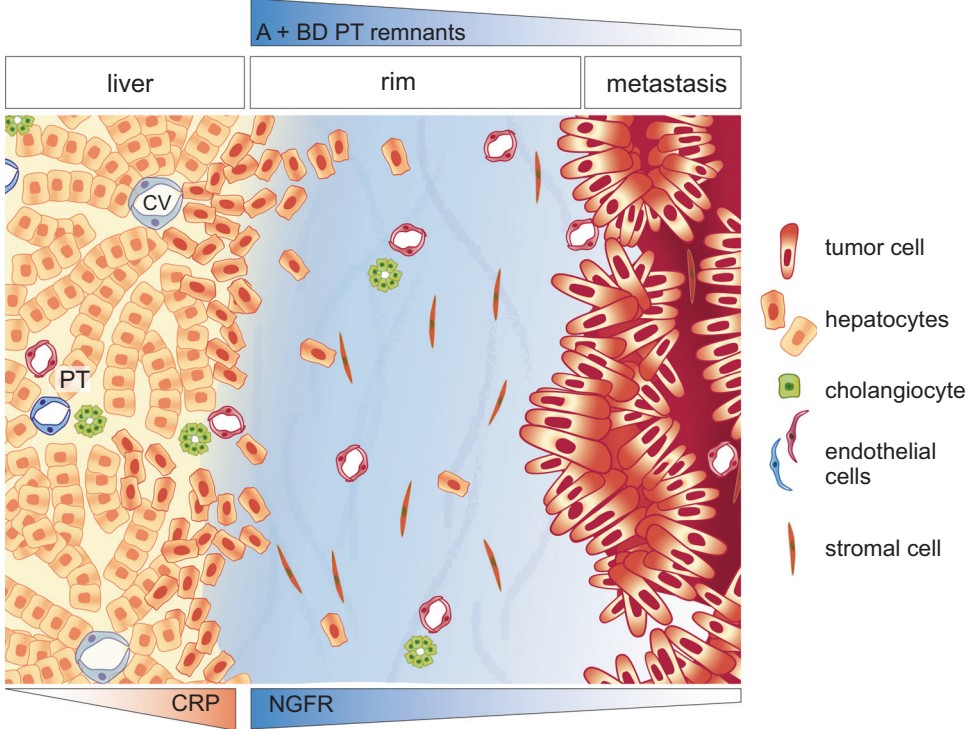

**Fig. 8 | Illustration of zonation of the perimetastatic capsule.** Normal liver to the left, a central vein (CV) and a portal triad (PT) are indicated. At the outer part of the rim, liver atrophy and fibrotic stroma similar to benign liver fibrosis can be seen. PT remnants consisting of artery and bile duct (A + BD) or only artery surrounded by remnant of portal stroma (A), as well as hepatocyte-derived Albumin⁺ cells can be found in the rim. The stroma in the tumor still contains those remnants, although at a lower frequency. Together, the histology of the rim and its clinical characteristics (potentially inducible by chemotherapy and associated with decreased tumor cell viability, in conjunction with the proportion-dependency of outcome) suggest a model in which the perimetastatic capsule evolves upon failed replacement-type growth and represents a reparative hepatic injury reaction.

encapsulation. Quantitative spatial analyses of the fibrotic rim suggest that encapsulation evolves from the liver parenchyma towards the metastasis, rather than vice versa.

A likely explanation for the previous difficulties in connecting the growth patterns to tumor biological traits[14] is their multifactorial nature. In a multifactorial model, the growth patterns represent a convergent phenotype of different cellular traits. Biologically, this can be translated into distinct functional requirements to achieve replacement-type growth, including invasion, migration into the liver parenchyma, and immune evasion. Based on our results, we propose that if the essential requirements for replacement-type growth are not met, impaired growth elicits a hepatic injury reaction; this becomes evident as a fibrotic rim when the rate of replacement growth is sufficiently hampered, permitting mature hepatic fibroinflammation to develop.

Several lines of evidence support this model: We show that the capsule is heterogeneous and represents the zonal evolution of distinct fibrosis features starting from the liver-rim interface and extending to the rim-tumor border. Its outer region is enriched in NGFR^high cells and contains the most well-preserved liver parenchymal remnants. NGFR represents a benign-like stromal cell phenotype that disappears with the loss of liver cells (hepatocytes, reactive ductular cells) and tumor cell dominance. In contrast, the inner rim, adjacent to the tumor, contains scarce liver parenchyma remnants and loses a benign-like stromal phenotype (Fig. 8 summarizes these findings). This suggests an evolutionary trajectory of the microenvironment from the outer zone toward the tumor center. These findings localize the zone of active formation of the capsule to the liver side, which in some cases is several hundred micrometers away from detectable tumor cells. They also imply that the capsule primarily develops independent of active juxtacrine or paracrine signals from tumor cells (as in the canonical concept of "desmoplasia"), but rather as the combined result of failed replacement growth and liver injury.

Our results provide evidence for growth pattern plasticity, mainly from replacement ("tumor wins") to encapsulation ("liver wins"). From a tumor evolutionary perspective, we propose that replacement growth is the default pattern of successful aggressive tumor invasion; this interpretation is supported by the observation that replacement-type growth is a characteristic of the majority of mouse models[2], including those used in this study, and by our finding that metastases that progress on chemotherapy are dominated by replacement growth. A recent study[15] has reported growth pattern scorings of samples from a randomized clinical trial, EORTC 40983, in which patients either received neoadjuvant chemotherapy or were operated up-front[41]. Despite a limited sample size (*n* = 30 treated vs n = 40 untreated patients), visual scorings in this unbiased setting revealed the encapsulated pattern to be significantly more frequent in the treated patients. Together with our results, the data provide evidence that encapsulation is facilitated by chemotherapy, likely by impacting tumor cell fitness, impairing the ability of replacement growth. Our experimental results from metastasis ablation after sensitizing tumor cells to DT support this interpretation, as they unveil all features of the reparative changes of the liver seen in human metastases.

Finally, this model also accounts for the fact that encapsulated tumors are seen in some patients who have not received neoadjuvant chemotherapy: Given a propitious balance of factors that favor a mature hepatic injury reaction *vs.* the tumor's ability for replacement growth, the latter stalls at a point where a sufficiently enveloping capsule can develop to prevent further replacement. Detailed studies on chemonaive, encapsulated metastases could unveil genetic or epigenetic features of cancer cells that fail at continuous replacement-type growth, leading to capsule development. An interesting open

question is if the emerging fibrotic rim might preclude further tumor invasion in clinical situations when tumor cell fitness is first impaired, and then regained, as is regularly the case during breaks in palliative chemotherapy. Studying this will require non-invasive methods to assess the growth patterns, which are under development[2].

Extended histological scoring requires multiple steps, including extensive sampling, time-consuming slide selection and annotations, and expert review. Given this complexity and the fact that systematic sampling is part of the clinical routine at our Pathology Department, we cannot conclusively determine which steps in our approach are driving the difference between our findings and previous data[2,13,14], which would have been desirable. It is possible that the annotation resolution or the high number of sections per tumor used here is needed to improve prognostic stratification. Nevertheless, our scoring approach is prohibitively time-consuming for clinical use. Developing automated image analysis tools may make fine-grained routine assessments of the growth patterns feasible in the future; standardized sampling, as suggested in the recent guidelines[2], will then be particularly important. Meanwhile, we endorse the dichotomic use of 100% encapsulation *vs.* all other fractions[2] to identify those patients with an excellent prognosis after resection as a clinically feasible approach. However, we would encourage using the term "encapsulated" rather than "desmoplastic" to denote the capsule, which avoids conflating a fibro-reparative reaction with cancer-cell-induced stroma formation.

Biologically, our results highlight the important role of the tumor-hepatocyte contact that defines the difference between replacement and encapsulated metastases, in driving metastasis invasion. Targeting their crosstalk could elicit a hepatic injury reaction, permit perimetastatic fibrosis, and thereby provide an unexplored avenue for targeted treatment of liver metastases.

## Methods

### Ethics statement
The Swedish National Ethical Review Board, *Etikprövningsmyndigheten*, gave ethical approval for all the work on human samples (#2019/01571 and #2021/06863-0, as well as #2018/1261-31 and 2019/05198); informed consent was waived. The participants received no compensation. The Swedish Board of Agriculture approved the animal experiments via the regional ethics committee, *Linköpings djurförsöksetiska nämnd* (#217-2022 and # 22149-2022).

### Patients
For the main cohort, all consecutive patients 18 years of age or older with CRLM operated on between 2012 and 2015 were identified in electronic databases at Karolinska University Hospital, Huddinge, Sweden. For the extended cohort that served to identify patients who progressed on chemotherapy, had a diagnosis of pre-operative liver fibrosis, or were not operated on, electronic charts were accessed and reviewed for the time frames indicated in the main text. All patients included in the main cohort underwent surgical resection of one or more CRLM at Karolinska University Hospital. Previous or synchronous diagnosis of colorectal cancer was confirmed histologically and on a multidisciplinary tumor board. Clinical data were collected by retrospective review of the electronic patient records. Data on tumor mutations (such as *KRAS*, *BRAF*, *NRAS*, and MSI status) were extracted from routine clinical analyses. "Synchronous" metastases were defined as metastases diagnosed within three months of diagnosis of the primary tumor. "Neoadjuvant treatment" was defined as chemotherapy that was given within three months prior to metastasis surgery. "Right-sided" was defined as cecum, ascending colon, hepatic flexure, and proximal two-thirds of the transverse colon. "Left-sided" was defined as the distal third of the transverse colon, the splenic flexure, sigmoid colon, descending colon, and the rectum. In case the precise location of a tumor in the transverse colon could not be determined, this tumor was considered right-sided.

For IHC stains, a series of $n = 6$ CRLM patients ($n = 2$ males and $n = 4$ females) was included. Of these, $n = 3$ had metastases that originated from left-sided primary colorectal adenocarcinomas, $n = 3$ from right-sided, and $n = 3$ patients received neoadjuvant chemotherapy. In $n = 5$ patients, margin-free tumor resection (pR0) was achieved. Tumors from $n = 3$ patients had confirmed *KRAS* mutations, two were *KRAS* wild type (one of which had an NRAS mutation), and for one, the *KRAS/BRAF/NRAS* mutation status was unknown. MSI status was unknown for all six tumors. For RNA ISH against *ALB*, four randomly selected metastases from these six patients were used. For multiplex-IF, material from $n = 6$ patients with predominantly desmoplastic metastases was selected, not overlapping with the IHC cases. An additional, non-overlapping two patients with predominantly encapsulated tumors were included in the multiplex-ISH analyses.

Patients with benign fibrotic conditions and patients with cholangiocarcinoma, and liver metastases from other primary tumors were identified by the authors (CFM and BB) during routine clinical diagnosis and by searching the electronic pathology database at Karolinska University Hospital. Images are representative of histopathological findings in these cases and the numbers of patients included for each condition are given in the Figure legends. Antibody stains for these patients were part of the clinical routine.

Slide selection, staining, digital annotations, and curation were done blinded to clinical outcome and treatment such as neoadjuvant therapy. Clinical data were collected without knowledge of growth pattern annotation results.

### Slide selection to approximate a panoramic central slice
Hematoxylin & eosin slides for each probe (liver resection specimen) were retrieved from the pathological archive and reviewed for each individual metastasis with an optical microscope. The slides approximating a panoramic central slice along the largest tumor diameter were selected for each individual metastasis. In cases where the growth pattern in non-central slices significantly differed from the pattern in the selected slides (i.e., peripheral and central slices of a particular metastasis had different patterns), the former were also included. For consistency and quality in the selection process, all slides were reviewed by one experienced liver pathologist (CFM). Slides were digitized using a Hamamatsu NanoZoomer S360 digital slide scanner at ×40 magnification.

### Digital annotation of the growth patterns
To annotate the GPs, the Hamamatsu software NDP.view 2, version 2.7.25, was used. Briefly, the invasion front of each slide was reviewed to confirm the quality of the WSIs before evaluating and annotating the entire liver-tumor interface. For each growth pattern previously described ("desmoplastic/encapsulated", "replacement type 1", "replacement type 2" and "pushing")[4], a specific color and label was assigned. Replacement types 1 and 2 were later combined as "replacement", in line with the latest consensus guidelines[2], and because of the difficulty in reproducibly differentiating them. Tumors with complete pathological regression were excluded from the analysis, following the scoring guidelines[2].

Using the freehand line tool in NDP.view 2, the liver-tumor interface was manually annotated for each WSI; adhering to scoring guidelines, portal zones and necrosis were not annotated, as they are considered to not represent the liver-tumor interface[2]. In addition, the percentage of viable tumor cells within the tumor was estimated visually for each WSI.

The percentage of each growth pattern per slide was calculated by dividing the total annotation length for the respective growth pattern by the sum of the length of all annotations on the slide and multiplying the result by 100. Accordingly, the percentage of each growth pattern per tumor or patient was calculated by dividing the sum of the length of the specific growth pattern annotation on all slides from the tumor

or patient by the sum of the length of all annotations from the tumor or patient and multiplying the result by 100.

## Statistics and reproducibility

Data were analyzed in R, version 2022.02.3. The clinical table was built using gtsummary, version 1.6.1, and finalfit, version 1.0.4. Data analysis and visualization was done with dplyr, version 1.1.2, readr, version 2.1.4, caTools, version 1.18.2, Rmisc, version 1.5.1, gplots, version 3.1.3, tidyr, version 1.3.1, and ggplot2, version 3.3.6, as part of tidyverse, version 1.3.1. Survival analyses, both uni- and multivariate, were performed with survival and survminer, version 3.3-1 and 0.4.9, respectively. No imputation of missing data was done. No statistical method was used to predetermine sample size for the retrospective studies of human data. For the analyses of clinical data specifically, all consecutive patients were included. For growth pattern analysis, all patients with available histological sections and who had no complete regression (no viable tumor cells) were scored. For mouse experiments, prior estimation of the sample size was done using pnorm in R, assuming a mean of 95% (standard deviation 5%) replacement in vehicle-treated mice and a mean of 80% replacement (standard deviation 15%) in treated mice (DT or chemotherapy) and a sampling ratio of 1, yielding a sample size of $n = 7$ with a power of 0.8 and a type 1 error rate of 5%. In total for each experiment, $n = 30$ mice were included for tumor injection to account for tumor engraftment failure and errors in the assumptions. All mice with visible tumors on follow-up ultrasound were included for treatments. In the studies on the human samples, the investigators were blinded to clinical outcome and treatment when scoring the growth patterns. In the mouse experiments, the investigators were blinded to the treatment group when scoring the growth patterns and tumor surface area. The investigators were not blinded to the treatment group during injections and when harvesting the tissue. Randomization was done as described below ("*Ultrasound-guided injection, treatment of mice, and tissue processing*"). In case data are presented as Box-and-Whisker plots, the median (line), the inter-quartile range (IQR, box), minimum and maximum values within 1.5 times the IQR from the first and third quartiles (whiskers), and the individual datapoints are shown. All statistical tests were two-sided. Sex was considered as a variable in univariate analysis, but was not included in multivariate analysis because it remained non-significant in univariate analyses. Sex was significantly different between patients who received or did not receive neoadjuvant chemotherapy; the potential reasons for these differences were not explored further in this study.

## RNA in situ hybridization (ISH)

RNA ISH was performed on 4–5 μm sections of formalin-fixed, paraffin-embedded patient samples using the RNAscope Multiplex Fluorescent Reagent Kit v2 (ACD, catalog no. 323100) and the RNAscope HiPlex12 Reagent Kit (488, 550, 650, 750) v2 (ACD, catalog no. 324409), according to the manufacturer's instructions. The probes Hs-Alb (600941), Hs-COL1A1-T7 (401891-T7), Hs-DCN-T6 (589521-T6), Hs-THY1-T3 (430611-T3), Hs-FN1-T1 (310311-T1), Hs-PDGFRA-T2 (604481-T2) and Hs-SPP1-T5 (420101-T5) were used. Samples were baked at 60 °C for 1 h, deparaffinized, and rehydrated followed by manual antigen target retrieval. Standard tissue pretreatment conditions were applied. For the multiplex reagent kit, Tyramide Signal Amplification (TSA®) Plus Fluorophores were diluted in RNAscope® Multiplex TSA buffer (ACD, catalog no. 322809) as follows: Fluorescein (Perkin Elmer, FP1168015UG diluted 1:1500) and Cy5 (Perkin Elmer, FP1171024UG, diluted 1:1000). Nuclear counterstain was performed with 4,6-diami-dino-2-phenylindole (DAPI) immediately followed by mounting with ProLong Gold Antifade (P36930). Images were captured with a Prime 95B sCMOS (multiplex reagent kit) or a Kinetix sCMOS (HiPlex reagent kit) photometrics camera on a Nikon Eclipse Ti2 inverted confocal microscope with a Crest V3 spinning disk. For Albumin detection a z

stack in 0.9 μm steps covering a total z range of 5 μm on a 2138.64 μm × 1527.6 μm field of view was captured. ImageJ, version 2.0.0, was used to generate maximum intensity projections of stacked images. For sections stained with the HiPlex reagent kit, images were captured in widefield mode.

## Immunohistochemistry

Antibodies and specific staining conditions for IHC are specified in Supplementary Table 3. Briefly, formalin-fixed, paraffin-embedded (FFPE) tissue samples derived from routine pathological diagnostics of CRLM were cut at 4–5 μm thickness. Stains were performed on a Leica (Germany) BOND-MAX automated staining machine at Karolinska University Hospital, Huddinge, Sweden. Pretreatment was performed with Bond Epitope Retrieval Solution 2 EDTA (Leica) for 20 min. m-IHC stained slides were quantified as specified below. Manual immuno-fluorescence staining for alpha-smooth muscle actin (ASMA) was performed on 4–5 μm tissue sections from murine FFPE samples. Sections were incubated at 60 °C for 1 h, deparaffinized, and rehydrated. Heat-induced antigen retrieval (HIER) was done in DIVA Decloaker buffer (DV2004MX, Biocare Medical) in a 2100 Antigen Retriever (Aptum Biologics Ltd). Sections were blocked with 1% BSA (A7960-100G, Sigma Aldrich) and 10% goat-serum (G9023, Sigma Aldrich) in TBS with 0.05% Tween 20 and 0.1% Triton X-100 for 1 h at room temperature. Tissue sections were stained with anti-alpha smooth muscle actin antibody (Abcam, Cat# ab5694, RRID: AB_2223021, 1:200) over night at 4 °C. Sections were washed in TBS with 0.05% Tween 20 and incubated with Alexa Fluor 647-conjugated secondary antibody (Thermo Fisher Scientific, Cat# A21245, RRID: AB_2535813, 1:400) for 1 h at room temperature, counterstained with DAPI and mounted in Aqua-Poly-Mount. Fluorescence imaging was done on a Leica LSM 710 confocal microscope.

## Multiplex-immunofluorescence

The multiplex-immunofluorescence (mIF) stainings were done either manually or using a semi-automated stainer. The results were compared and combined for analysis. Semi-automated staining was performed on the Bond RX$_m$ autostainer (Leica Biosystems). Deparaffinization was done with Dewax solution (Leica Bond, Cat# AR9222) followed by an initial antigen retrieval with Epitope retrieval solution 2 (Leica Bond ER2, Cat# AR9640) at 95 °C for 30 min. Six target staining cycles were performed in the following order: anti-LOXL2 (clone E3P7Y, Cell Signaling Technology, Cat# 99680; 1:150), anti-ASMA (clone 1A4, Dako, Agilent Technologies, Cat# M0851, RRID:AB_2223500; 1:350), anti-PDGFRa (clone D13C6, Cell Signaling Technology, Cat# 5241, RRID:AB_10692773; 1:250), anti-PDGFRb (clone 28E1, Cell Signaling Technology, Cat# 3169, RRID:AB_2162497; 1:150), anti-FAP (clone E1V9V, Cell Signaling Technology Cat# 66562, RRID:AB_290419; 1:500) and anti-pSTAT3 (clone D3A7, cell Signaling Technology, Cat# 9145, RRID:AB_2491009; 1:150). Each cycle included a blocking step with Dual Endogenous Enzyme Block (Agilent Technologies, Cat# S2003) for 10 min at RT, primary antibody incubation for 30 min at RT, incubation with secondary ImmPRESS-mouse HRP (Vector Laboratories, Cat# MP-7402,) or ImmPRESS-rabbit HRP (Vector Laboratories, Cat# MP-7401) respectively for 10 min at RT before fluorescent labeling by a tyramide signal amplification step with Opal dyes 570, 480, 520, 620, 290 and TSA-DIG/780 (all Akoya Biosciences), diluted 1:150 in 1X Plus Automation Amplification Diluent (Akoya Biosciences, Cat# FP1609) for 10 min at RT (except for Opal 480, 1:200 dilution). Each staining cycle was completed by an antigen retrieval step with Epitope retrieval solution 1 (Leica Bond ER1, Cat# AR9961) at 95 °C for 20 min. A final DAPI staining for 5 min at RT (Akoya Biosciences, Cat# FP1490) was performed before mounting with ProLong Diamond Antifade media (Thermo Fisher Scientific, Cat# P36970). All staining steps had subsequent washing steps with Wash solution (Leica Bond, Cat# AR9590).

Manual staining was performed with the Opal 6-Plex manual detection kit (Akoya Biosciences, Cat# NEL861001KT) following the manufacturer's instructions with minor modifications. Briefly, tissue sections were deparaffinized, rehydrated, rinsed in distilled water and subjected to microwave heat-induced epitope retrieval (HIER) at pH 9 (Dako, Agilent Technologies, Cat# S2367) for 15 min. After blocking with 2.5% normal horse serum (Vector Laboratories) for 10 min at RT, sections were incubated with the respective primary/secondary antibodies of the mIF panel and Opal reagents (Akoya Biosciences). Washes between incubations were performed in Tris-buffered saline with 0.1% Tween 20 (TBST). The mIF panel consisted of antibodies for the detection of FAP (clone EPR20021, Abcam, Cat# ab207178, RRID:AB_2864720; 1:700) with Opal 480, PDGFRa (clone D13C6, Cell Signaling Technology, Cat# 5241, RRID:AB_10692773; 1:100) with Opal 570, PDGFRb (clone 28E1, Cell Signaling Technology, Cat# 3169, RRID:AB_2162497; 1:100) with Opal 520, ASMA (clone 1A4, Dako, Agilent Technologies, Cat# M0851, RRID:AB_2223500; 1:200) with Opal 620 and tissue factor (TF, Atlas Antibodies, Cat# HPA049292, RRID:AB_2680701; 1:200) with Opal TSA-DIG/780. ImmPRESS HRP horse anti-mouse IgG (Vector Laboratories, Cat# MP-7402) and anti-rabbit IgG (Vector Laboratories, Cat# MP-7401) were used as secondary antibodies. Microwave HIER at pH 6 for 15 min was performed after every staining cycle. Nuclear counterstaining was achieved using DAPI. Tissue sections were mounted with ProLong Diamond Antifade media (Thermo Fisher Scientific, Cat# P36970) and imaged using the PhenoImager HT system (Akoya Biosciences) and the MOTIF whole-slide multispectral mode at a resolution of 0.5 µm/pixel. Staining quantification was done in QuPath (v. 0.2.3) as described separately.

### Mice
C57BL/6J mice obtained from Charles River were used for all experiments. Female and male mice at 9–12 weeks of age were included. Mice were housed in specific-pathogen-free conditions at a 12 h light/dark cycle at circa 20–22 °C and fed standard chow.

### Cell culture
A pancreatic cancer cell line was derived from the KPCT mouse model in-house by culturing dissociated, minced pieces of a KPCT adenocarcinoma and provided by Rainer Heuchel; cells were derived from KPC (KrasLSL-G12D/+;Trp53LSL-R172H/+;Pdx-Cre) mice[42] that had been bred to B6.Cg-Gt(ROSA)26Sortm9(CAG-tdTomato)Hze/J mice to generate KPCT mice. MC38 cells were obtained from Kerafast, USA (Cat# ENH204-FP, a gift from James W. Hodge), and adenocarcinoma cell identity was histologically confirmed after cell injection into the liver. Cells were cultured in DMEM/F12 medium (Gibco) with 10% FBS (Sigma-Aldrich) and 1% Penicillin-Streptomycin (Sigma-Aldrich) at 37 °C in 5% CO$_2$. At the timepoint of injection, MC38 cells were in passage 3 after thawing, and KPCT-DTR cells in passage 25–30.

### Generating diphtheria toxin receptor-bearing tumor cells
To create lentiviral particles expressing DTR-GFP, a DTR-GFP fragment was obtained by PCR from plasmid pAAV-FLEX-DTR-GFP (Addgene #124364, a gift from Eiman Azim & Thomas Jessell) using forward primer, cttccatttcaggtgtcgtgacCTGCAGGAATTCGCCACCATGAAG, and reverse primer, ttaccgataagcttgatatcGAATTCCGGCCGCTTTACTTGTACAGCT. The resulting PCR fragment was cloned by Gibson assembly into BsiWI/EcoRI digested lenti.Cas9.BFP.Blast (Addgene #196714) and verified by Sanger sequencing. The resulting plasmid lenti.DTR.GFP (Addgene #201062) was packaged into lentiviral particles in HEK-293T (ATCC) with packaging plasmids psPAX2 (a gift from Didier Trono, Addgene #12260) and pCMV-VSV-G (a gift from Bob Weinberg, Addgene #8454). KPCT cells were transduced with lentivirus, and DTR-GFP expressing cells were enriched to purity by repeated sorting for GFP-positive cells (FACS instrument Sony SH800). The

ability to induce cell death in transduced KPCT cells (KPCT-DTR) was tested in vitro by incubating the cells for 24 h with diphtheria toxin (Sigma-Aldrich, # 322326) in concentrations ranging from 10 µg/mL to 5 pg/mL. The impact on metabolic activity was quantified by the Cell Proliferation Kit I (MTT) (Roche # 11 465 007 001). Non-transduced KPCT cells served as reference.

### Ultrasound-guided injection, treatment of mice, and tissue processing
The Vevo 3100 preclinical imaging system (Visualsonics, Toronto, Canada) was used for tumor cell injection and follow-up of tumor growth. Briefly, hair was removed from the animals over the abdomen and the mice were anesthetized using isoflurane inhalation. Mice were scanned in a supine position on a pre-warmed table. The liver was identified morphologically and in the case of tumor cell injection, $10^5$ cells were injected with a 30 Gauge injection needle, penetrating the skin through the left upper quadrant. Tumor formation was confirmed by liver ultrasound on day 15 after injection, and it was decided a priori that only mice with visible tumors on ultrasound on d15 are included in downstream analysis. Seventeen KPCT-DTR tumor-bearing mice were treated either with PBS ($n = 8$) or diphtheria toxin (DT, $n = 9$). For treatment group assignment, mice were divided into two blocks based on their sex and in-block randomization was performed with the RAND() function in Microsoft Excel. The researcher performing the treatments (NG) was aware of group allocations. Mice received DT at 0.01 mg/kg body weight (Sigma-Aldrich, # 322326) or PBS by intraperitoneal injection on days 15 and 19, and they were sacrificed on day 23 ($n = 3$ mice of each group) or day 26 (all remaining mice). For the chemotherapy study, twenty-four MC38 tumor-bearing mice were included and randomized into chemotherapy ($n = 14$) or saline ($n = 10$) treatment group as described. In one mouse of the treatment group, no microscopic tumor was seen after processing and no growth pattern could be analyzed. Mice received either 6 mg/kg Oxaliplatin (Abcam, #ab141054) in combination with 50 mg/kg 5-Fluorouracil (Bio-Techne, # 3257/50) or saline by intraperitoneal injection on days 15 and 19. Mice in the chemotherapy group were injected with 0.05 mg/kg buprenorphine (Temgesic, #521634, Indivior) intraperitoneally before treatment to ameliorate pain from peritoneal irritation. All mice were sacrificed on day 24 after injection. Mice were euthanized, liver tissue was harvested, transferred to 4% formaldehyde solution (#1004965000, Sigma Aldrich) for 24 h incubation at room temperature, and transferred to 70% EtOH for storage at 4 °C. Liver lobes were separated and the left and median lobes were imaged from both sides with an iPhone 11 camera. Tumor surface size was manually annotated and measured in ImageJ and these analyses were not blinded. The tumor surface area is the sum of the areas of all visible tumor lesions for each mouse. After imaging, the tissue was embedded in paraffin and sectioned at 4–5 µm thickness, such that the largest diameter of the visible tumor was captured on the slide, and slides were scanned. For each animal, one whole slide image of a hematoxylin and eosin-stained tissue section was used for blinded growth pattern and tumor cell viability scoring; for the experiment with DT specifically, complete regressions were seen, such that a clear intrahepatic lesion but no viable tumor was identified in the lobe in which tumor cells had been injected and for which growth had been confirmed with ultrasound. The growth patterns in all the murine metastases, including individual regressed lesions, were scored independently by two raters (CFM and MG) and the mean of both assessments was used to compare the groups. A maximum tumor volume was not defined by the ethics committee; the humane endpoint was defined using a combined score including the animals' weight, body posture, physical appearance as well as urine and fecal excretions.

## Staining quantifications

Protein or RNA expression was quantified in the "desmoplastic rim" (DR), the perimetastatic liver parenchyma (PLP), and the tumor center (TC). WSIs of immunohistochemistry and multiplex immuno-fluorescence staining were quantified in QuPath[43]. RNA ISH was quantified based on scans from selected regions of interest, such that three and five regions of interest, respectively, were imaged covering a 5 mm × 5 mm field of view per region. The DR was annotated manually and defined as the area dominated by stromal cells facing the metastasis-liver interface outwards and tumor cells and necrosis inwards, thus yielding rims with different widths. Next, the DR annotations were expanded at a fixed distance of 1500 μm (IHC/IF) or 1000 μm (ISH) into the perimetastatic liver parenchyma. Portal tracts were manually annotated in the DR and PLP, subtracted from the DR and PLP annotations, and excluded from the quantification (Supplementary Fig. 11). Cases with extensive necrosis were excluded from the quantification for the TC. For one case in the IHC quantification, a trained pixel classifier was used to separate the stroma from the tumor tissue and to create tumor and stroma annotations due to complex and labor-intensive tumor/stroma separation, while the remaining cases were annotated manually. Next, stain vectors for color deconvolution of chromogenic stains were determined in QuPath using the built-in tool for 3,3'-Diaminobenzidine (DAB) and hematoxylin and by selecting an area of pure staining for alkaline phosphatase (AP). Next, the obtained stain vectors were applied to the chromogenic images. For the quantification of fluorescent images, the signal intensities of each target were analyzed in separate channels.

Next, each annotation, DR, PLP, and TC, was divided into 25 μm × 25 μm tiles (Supplementary Fig. 11). The mean intensities for each marker and the distances to the invasion front were automatically calculated for each tile in QuPath. Raw measurements were exported in tabular format for analysis.

## Analysis of spatial annotations

Protein expression for each case and marker was analyzed in R. Because the thickness of the DR and TC varied for each case, the distances were normalized–including the PLP for consistency–such that the maximum width (=maximum distance) of each region corresponded to "100%" using the following equation: Standardized distance = (100 / maximum distance * tile distance [from the liver-tumor interface for the PLP and DR regions; or from the DR for the TC region]). Considering the liver-tumor interface as a distance of 0%, the entire widths of the PLP, DR, and TC regions were denoted, for example, as 100%, −100%, and −200%, respectively, and the standardized distances were translated accordingly. Next, for each case and marker, the mean tile intensities were averaged over cut-off distances of 5%. Finally, the intensities were averaged for the whole series and smoothed using a rolling window of width = 5 distance intervals.

## Identification and annotation of the intratumoral portal triads

All cases with an overall predominantly encapsulated pattern (>95%, n = 55 patients) were selected for annotation of portal tracts. Since portal veins are most often obliterated, remnants of portal tracts were quantified either when hepatic arteries and bile ducts were seen together (artery and bile duct, "A + B"") or when presented as isolated arteries surrounded by remnants of portal stroma (artery, "A"). Subsequently, all occurrences of A + BD and A throughout the tumor nodule were annotated on NDP.view 2 (U12388-01) using the "pin" tool. The DR area was annotated using the "freehand region" tool. For spatial quantification, two further annotations were added manually: an outer and inner border of the DR corresponding to the liver-rim interface and the rim-tumor interface, respectively. Next, all annotations were transferred from NDP.view 2 to QuPath using a script written in Groovy. The minimal distances from each pin to both the liver-rim and the rim-tumor interface were calculated, and the data were exported from QuPath in tabular format for subsequent analysis in R. The relative distance of each portal tract annotation to the liver-rim interface was finally calculated.

## Reporting summary

Further information on research design is available in the Nature Portfolio Reporting Summary linked to this article.

## Data availability

The experimental data generated in this study are provided in the Figures, Supplementary Information and in the Source Data file. The processed clinical data generated from the patient cohort are available under restricted access for privacy and legal issues. Access can be obtained for research purposes by submitting a request to the corresponding author (M.G.) by email, specifying contact details, affiliation, and purpose of the request. All requests for clinical data used in this study are reviewed by the principle investigators responsible for the clinical cohort (M.G., C.F.M., and J.E.), and requests will be answered within four weeks. Any data that can be shared after review by the National Swedish Ethical Review Board will be released via a Data Transfer Agreement to the requesting party; this includes all relevant pseudonymized clinical data (such as relapse and overall survival, information on neoadjuvant treatment, as well as the digitally annotated imaging data, including hematoxylin & eosin stains and immunohistochemistry stains with annotations, used to generate the growth pattern quantifications and protein profiling). Authorship requirements for the use of the clinical data from this cohort will be regulated in a Collaboration Agreement, in which the authors agree on authorship requirements prior to data sharing. Source data are provided with this paper.

## Code availability

Publicly available packages were used to conduct analyses, as specified in Methods. The data processing and analysis code is available on GitHub (https://github.com/gerlingm/CRLM_analysis).

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

## Acknowledgements

This study was supported by The Swedish Research Council (project nr. 2018-02023), The Swedish Society for Medical Research, the Åke Wiberg Foundation, the Jeansson Foundation, The Swedish Cancer Society (22 2175 Pj), the Karolinska Institute (all to M.G.), and Cancer Research KI (to M.G. and E.S.). J.E. is supported by Region Stockholm and by the Bengt Ihre Foundation. N.G. is funded by the Deutsche Forschungsgemeinschaft (DFG, German Research Foundation, grant nr. 460567311), the Ruth and Richard Julin Foundation, and the Alex and Eva Wallström Foundation. C.F.M. is supported by The Swedish Society for Medical Research (PD21-0114). C.S. received research grants from the Swedish Cancer Society (21 0401FE, 21 1749Pj), the Swedish Research Council (2022-01151), and the Trond Mohn Foundation (TMS2022STG01). We are grateful for the support provided by the histological and immunohistochemical laboratories at the Department of Clinical Pathology and Cancer Diagnostics, Karolinska University Hospital, and Sólrún Kolbeinsdóttir for curating the code used for the analysis of clinical data. We gratefully acknowledge the support of Peter Bankhead for staining quantification with QuPath, and we thank Rune Toftgård and Nick Tobin for comments on the manuscript and valuable discussions. We are grateful for help from Ying Zhao, Preclinical Imagine Facility, Karolinska Institutet, as well as from Makbule Sagici and Poomy Pandey, Morphological Phenotype Analysis (FENO), Karolinska Institutet. Parts of

this study were performed at the Live Cell Imaging core facility/Nikon Center of Excellence at Karolinska Institutet, supported by the KI infrastructure council. The CRISPR Functional Genomics is funded by SciLifeLab.

## Author contributions

C.F.M. and M.G. designed and supervised the study, B.B. conceived the concept of perimetastatic injury, established the clinical multiplex immunohistochemistry stains and interpreted their results. E.T.Q., D.K., L.B., S.H., Y.H., and M.S.S. performed slide selection and manual annotations, supervised by CFM. B.B. selected and interpreted cases of benign fibrosis and cholangiocarcinoma and interpreted all findings. P.B.V. and L.D. performed visual growth pattern assessments, were involved in interpreting the data and commented on the manuscript. N.G. analyzed clinical data, performed confocal microscopy, and generated the plots for most figures. L.H. performed immunofluorescence and analyzed in situ hybridization data. R.H. generated the KPCT cells. C.S., A.L., A.M.B. and A.Ö. performed and interpreted multiplex immunofluorescence stains. N.G. and C.F.M. wrote the code for analyzing the annotation-derived and clinical data, N.G., C.F.M., S.H., and Y.H. wrote the code to analyze the spatial quantification of protein expression. J.E. and M.G. collected most clinical data; C.V. specifically collected data on patients who progressed on chemotherapy. J.E. interpreted clinical data and supervised the inclusion of clinical variables. E.S. interpreted the clinical data and commented on the manuscript. S.S. and N.G. performed in situ hybridization, interpreted the related data, prepared figures, and commented on the manuscript. N.G. and M.G. performed the animal experiments with the help of S.S. A.D.V. commented on the manuscript and interpreted the data. A.C.N., S.D., and B.S. created the KPCT DTR-EGFP expressing cell line. M.G. wrote the manuscript with input from C.F.M., B.B., N.G., and A.D.V.

## Funding

## Competing interests

The authors declare no competing interests.

## Additional information

[1]Department of Biosciences and Nutrition, Karolinska Institutet, 14183 Huddinge, Sweden. [2]Department of Clinical Pathology and Cancer Diagnostics, Karolinska University Hospital, Stockholm 14186, Sweden. [3]Department of Laboratory Medicine, Division of Pathology, Karolinska Institutet, 14186 Stockholm, Sweden. [4]Department of Clinical Genetics, Pathology and Molecular Diagnostics, Medicinsk Service, Skåne University Hospital, 22185 Lund, Sweden. [5]Department of Immunology, Genetics and Pathology, Uppsala University, 75185 Uppsala, Sweden. [6]Pancreatic Cancer Research Laboratory, Department of Clinical Science, Intervention and Technology, Karolinska Institutet, 14183 Hudinge, Sweden. [7]Department of Oncology-Pathology, Karolinska Institutet, 17176 Solna, Sweden. [8]CRISPR Functional Genomics, SciLifeLab and Department of Medical Biochemistry and Biophysics, Karolinska Institutet, 17165 Solna, Sweden. [9]Department of Medicine Huddinge, Karolinska Institutet, 14183 Huddinge, Sweden. [10]Department of Clinical Science, Intervention and Technology, Division of Surgery, Karolinska Institutet, Karolinska University Hospital, 14152 Stockholm, Sweden. [11]Translational Cancer Research Unit (GZA Hospitals and University of Antwerp), Antwerp, Belgium. [12]Centre for Cancer Biomarkers CCBIO, Department of Clinical Medicine, University of Bergen, 5020 Bergen, Norway. [13]Theme Cancer, Karolinska University Hospital, 17 176, Solna, Sweden. [14]These authors contributed equally: Carlos Fernández Moro, Natalie Geyer, Sara Harrizi, Yousra Hamidi. ✉e-mail: marco.gerling@ki.se

