## [Peer Review File · Nature Communications]

REVIEWER COMMENTS

Reviewer #1 (Remarks to the Author): expertise in liver metastasis histology

In their manuscript "An idiosyncratic, zonated stroma encapsulates desmoplastic liver metastases and originates from injured liver" by Moro et al. the authors use digitally annotated histopathological slides and different staining methods to investigate the origin and clinical implications of different growth patterns of colorectal liver metastases. They confirm that a desmoplastic growth pattern is associated with a better outcome and provide first evidence that the associated stroma is a reaction of the liver rather than originating from the tumor cells. The study is interesting and addresses an intriguing tumor biological question. It is mostly well written and the figures support the respective arguments. However there are a number of issues which need to be addressed to improve this study:

Abstract:

- Please state more clearly, what the rationale, the question / objective, the findings, and the conclusions of the study were.
- While 60.000 annotations sound impressive, please also provide the number of patients, which would be a more applicable parameter to determine the scope of the study.

Introduction:

- None.

Methods / Results:

- The numbering of the figures does not correspond to the numbering in the text. This makes it really frustrating to assess the manuscript, especially as the figures have been embedded in the text.
- In Figure 1 the preprocessing is nicely illustrated. How were the WSIs that belong to one tumor "puzzled" together? With a macroscopic photo? Automatically? Please describe.
- What was the rationale behind "only" measuring length of a certain growth pattern? One can imagine that a lot of information / accuracy might be lost due to this 1-dimensional measurement. Have the authors tried using an automated (AI / ML based) approach of detecting the different growth patterns? QuPath as well as other programs could be used to train a classifier (desmoplastic / tumor / liver) which could then be utilized to determine the predominant growth pattern. This would be more objective, less error prone and could easily be automated even for large numbers of WSIs.
- The authors state "[the] Boolean model [...] is difficult to reconcile with the probabilistic character of biological systems" (line 136 f) and argue that their approach better addresses the likely continuous nature of these changes. However, as far as I understand, all they have done is introduce another cut point at 33%. How were these thresholds / strata / cut points determined? Have the authors tried statistical methods to identify the optimal cut points (i. e. via ROC analysis or capabilities from the survival / survminer package in R, etc.)? I would be really interested in the Kaplan-Meier-curves, when 5 or more strata (0-20%, 21-40%, 41-60%, etc.) of desmoplastic growth are used. As there are two main groups (desmoplastic vs. replacement), this would mean that for the replacement type the correlation would be exactly inverse. Is this the case?
- The association of growth pattern with chemotherapeutic effects is interesting but needs to be explored in more detail. Clinically, the effect of chemotherapy on liver metastasis is usually scored using Rubbia-Brandt et al. (PMID: 17060484) for example. Have the authors data on the distribution of the growth pattern with respect to this response assessment? How do the authors explain that there is also a correlation between desmo GP and viable tumor cells in the chemo-naïve subgroup?
- For the argument that the desmoplastic stroma originates from liver damage and has less to do with the tumor cells themselves, it would be important to compare colorectal metastases to that of other primary tumors. Do the authors have any data on this? For example, from clinical experience, there are tumor types with a lower rate of desmoplastic stroma and better prognosis (e. g. neuroendocrine liver mets) and tumor types with higher rate of desmoplastic stroma and a worse prognosis (e. g. pancreatic liver mets). With the lack of functional experiments, more data would be needed to draw this conclusion.
- Another key factor to prove or disprove the author's theory would be to assess the function of

the surrounding liver. It would therefore be very important to include data on the function / appearance / morphology of the normal adjacent liver tissue. With hundreds of patients included in the study, there should be a subgroup of patients which has some form of liver pathology (i. e. fatty liver disease, fibrosis, cirrhosis, etc.). How is the distribution of desmoplastic / replacement pattern in these patients? Also, it would be really interesting to see the distribution of these growth patterns in tumors originating from the liver themselves (i. e. HCC) and the effect on clinical outcome.

- For the quantification of the marker expression (curves) please provide the respective standard deviations.
- The methodology of determining different types of portal triad destruction should be explained in better detail and supported by either data or literature. It is usually dependent on the type of liver damage and does not always follow the sequence vein, bile duct, artery. Please comment.

Discussion:

- Currently, the discussion is more of a repetition of the results section and needs to be revised. What is novel about the findings? What are the limitations / open questions? Etc.? To me it is unusual to refer to the figures in the discussion as well.
- The passage starting from line 409 to line 415 is somehow speculative as the authors have not provided any functional experiments in this regard. This should be framed differently.

Reviewer #2 (Remarks to the Author): expertise in bioinformatics of digital pathology

This is an interesting study addressing a clinically and biologically relevant topic. This study is the result of a lot of work, which the authors should be commended on. This involves a lot of work in sample collection, slide preparation, annotation and analysis.

However, there are two major flaws in the scientific reasoning.

The first major issue is the claim of causality. The whole study is descriptive and the authors cannot deduct that the desmoplastic reaction is caused by the liver instead of the tumor. This is something that you really need a controlled experiment for, typically a mouse experiment. The authors present an impressive amount of descriptive data on the morphology of the desmoplastic rim, and about gradients of molecules in the desmoplastic rim. But all of these are still descriptive and have several possible reasons - they are not proof of causality. Even Figure 6 is compatible with the tumor tissue inducing scarification of the surrounding liver tissue, as opposed to scarification induced by peritumoral liver tissue. The authors could fix this issue by removing claims of causality from this descriptive study, but this reduces the novelty.

The second major issue is the presence of selection biases in the cohort, which casts doubt on the claim that chemotherapy induces desmoplasia. The authors find that patients after neoadjuvant chemotherapy have more desmoplastic growth than chemo-naïve patients. However, this is highly biased due to two types of selection bias. (1) Oncologists and surgeons may be more likely to prescribe neoadjuvant therapy if the tumor seems slowly growing, as opposed to "out of control", even at the same absolute size of tumors. Hence, more favorable biological patterns can be enriched in the population which receives neoadjuvant therapy. (2) Even more gravely, neoadjuvant therapy selects the patient population for good prognosis patients. Patients who progress under therapy are less likely to be operated. Hence, the survivors of 2-3 months of chemo are enriched for favorable-biology tumors, i.e. desmoplastic mets. The authors discuss this in line 245-250, but I do not find that this argument resolves the bias.

Minor issues:

The authors mention "60k annotations" in their abstract, but this information is not helpful. What the reader is interested in is the number of tissue samples, and the number of patients. You can even generate 60k annotations in a single patient. Please state the patient number in the abstract.

The study should adhere to the STARD/TRIPOD or other relevant guidelines by the Equator

network.

Reviewer #3 (Remarks to the Author): expertise in molecular biology of colorectal cancer liver metastasis

The manuscript by Moro et al investigates the formation of two distinct type of colorectal cancer (CRC) liver metastases, which are referred to as desmoplastic and replacement type lesions. Patients with desmoplastic CRC liver metastases have better outcomes when compared to those with replacement type lesions. Previous studies suggested that only patients with 100% desmoplastic features exhibited better outcomes; however, if any portion of the metastases were replacement, the outcome was worse. The current manuscript argues that the proportion of desmoplastic features has prognostic significance. The authors perform detailed histological characterization of the desmoplastic ring from patient-derived CRC liver metastasis samples and put forth a model in which the stromal deposition is the result of an injury response in the liver, that increases deposition of the encapsulating stroma, rather than tumor-induced activation of cancer-associated fibroblasts/stellate cells in the liver.

The authors are investigating an important question regarding the formation of desmoplastic CRC liver metastases, which is an understudied and important area of research. The data presented in the manuscript comes from a careful characterization of desmoplastic lesions using an excellent resource of clinical material obtained from patients with CRC liver metastases. the authors are to be commended for the efforts invested to assess the degree of the desmoplastic ring around the entire circumference of the metastatic lesion. While the authors interpretation that desmoplastic lesions form as the result of a liver injury response that encapsulates the lesion and prevents parenchymal invasion, it would be important for the authors to test some of the predictions they make using pre-clinical mouse models. These are discussed in detail below.

Major comments:

- 1) The authors will need to carefully review the figure legends to ensure that the figures are labeled correctly. See notes under minor comments for specific examples of the disconnect between the text and the figure legends.
- 2) The authors argue that the cells present at the desmoplastic rim/liver interface are more similar to activated fibroblasts that are seen in the context of a wound-healing, fibrotic response. In contrast, the fibroblasts located close to the desmoplastic rim/cancer interface would be more reminiscent of cancer-associated fibroblast phenotype. It would strengthen the paper considerably if the authors could generate some additional data that would more clearly define these distinct fibroblast cell types. This could include spatial transcriptomic approaches or laser capture microdissection of alphaSMA+/NGFR+ cells (outer portion of the desmoplastic ring – adjacent to the liver) versus]alphaSMA+/NGFR- cells (inner portion of the desmoplastic ring – adjacent to the metastatic lesion). Alternatively, could fibroblasts/stellate cells be isolated from patient material and separated into NGFR+ and NGFR- cells for RNA sequencing.
- 3) The authors argue that a tissue response that leads to fibrosis might affect the type of lesion that develops in patients with CRC liver metastases. If this is the case, would the authors expect that patients with a previous history or current condition associated with liver fibrosis (Cirrhosis) would be more susceptible to the development of desmoplastic lesions? Has such an association been investigated? Could the current patient cohort developed by the authors be used to investigate this?
- 4) The authors suggest that the replacement type CRC liver metastases are the “default” type of lesion, and that desmoplastic lesions form secondary to a tissue repair/fibrosis response that encapsulates the lesion. Could this be tested in an animal model? Several CRC cell lines exist that form replacement type metastases when splenically injected (MC38 cells are an example). How would the formation of MC38-derived liver metastases be altered if these cells where injected into mice with active liver fibrosis (PMID: 34028753)? Would the default replacement type lesion

normally formed by MC38 cells in normal liver be shifted to a desmoplastic lesion when injected into a fibrotic liver? Would you see the same distribution of NGFR+/alphaSMA+ and NGFR-/alphaSMA+ fibroblasts/stellate cells in such a model?

Minor comments:

It is unusual to reference a manuscript figure in the introduction. Maybe a paper citation can be used when describing the features of the desmoplastic and replacement lesions in the introduction and the figure can be cited in the results.

Care should be taken with the Figure legends:

Page 13 (PDF): Figure Legend should indicate Figure 2 (not Figure 1)

Page 14 (PDF): Figure Legend should indicate Figure 3 (not Figure 2)

Page 19 (PDF): Figure Legend should indicate Figure 4 (not Figure 3)

Page 22 (PDF): Figure Legend should indicate Figure 5 (not Figure 4)

Page 24 (PDF): Figure Legend should indicate Figure 6 (not Figure 5)

Page 28 (PDF): Figure Legend should indicate Figure 7 (not Figure 6)

Page 17/18 (PDF): References to Supplementary Fig. 3 in the text should be referring to Supplementary Fig. 4

RESPONSE TO REVIEWERS' COMMENTS

We would like to thank the reviewers for their constructive feedback on our manuscript. We here provide a summary of the most important changes, followed by point-to-point answers.

The reviewers have raised important points regarding functional studies to substantiate our main claim that the development of a perimetastatic capsule is facilitated when cancer cell invasion of the hepatic plates is impaired.

In the revised manuscript, we address this point by providing data from mouse models of liver metastases. We treated mice in a colorectal cancer liver metastasis model with chemotherapy and observed perimetastatic encapsulation. We also developed a novel model that allows tumor cell ablation based on a diphtheria toxin system. Upon ablation of established metastases, we observed the complete spectrum of the reparative hepatic fibrosis response that characterizes human encapsulated metastases.

Our data provide evidence for a switch from replacement-type growth to encapsulation upon impaired tumor viability, which, to our knowledge, is the first functional proof of the plasticity of liver metastases growth patterns.

In addition to the functional studies, in the revised manuscript, we provide novel human data. Briefly, we demonstrate that colorectal cancer liver metastases that progress on chemotherapy show non-encapsulated growth. We address the potential selection bias in the sub-analysis of growth pattern fractions after chemotherapy using these new data and the *in vivo* studies. Furthermore, we provide a more detailed discussion of recent data on the growth patterns derived from analyzing samples from a randomized-controlled clinical trial.

To more clearly convey our main hypothesis, which is that failed metastasis invasion is linked to the development of the capsule, we revised the manuscript's structure and rewrote large parts of the abstract, the introduction, and the discussion.

Finally, as the reviewers pointed out, there was a mistake in the numbering of the Figures in our initial submission. We sincerely apologize for that oversight, which must have made the review process unnecessarily difficult.

We thank the reviewers for the careful assessment of our manuscript and we look forward to the comments on the changes and novel data.

Sincerely,

Marco Gerling (on behalf of all the authors)

The point-by-point answers to the reviewers' comments are given below (**referee comments in bold**, *our answer in italics*).

Reviewer #1 (Remarks to the Author): expertise in liver metastasis histology.

In their manuscript “An idiosyncratic, zonated stroma encapsulates desmoplastic liver metastases and originates from injured liver” by Moro et al. the authors use digitally annotated histopathological slides and different staining methods to investigate the origin and clinical implications of different growth patterns of colorectal liver metastases. They confirm that a desmoplastic growth pattern is associated with a better outcome and provide first evidence that the associated stroma is a reaction of the liver rather than originating from the tumor cells.

The study is interesting and addresses an intriguing tumor biological question. It is mostly well written and the figures support the respective arguments. However there are a number of issues which need to be addressed to improve this study:

We thank the reviewer for helpful suggestions on our manuscript.

Abstract:

• Please state more clearly, what the rationale, the question / objective, the findings, and the conclusions of the study were.

We have revised the abstract to better reflect this structure, while at the same time keeping it non-technical as suggested by the journal.

• While 60.000 annotations sound impressive, please also provide the number of patients, which would be a more applicable parameter to determine the scope of the study.

We have added the number of patients and removed the number of annotations following also a recommendation by reviewer #2.

Introduction:

• None.

Methods / Results:

• The numbering of the figures does not correspond to the numbering in the text. This makes it really frustrating to assess the manuscript, especially as the figures have been embedded in the text.

We sincerely apologize for this mistake and would like to thank all the reviewers for working their way through our manuscript despite this oversight.

• In Figure 1 the preprocessing is nicely illustrated. How were the WSIs that belong to one tumor “puzzled” together? With a macroscopic photo? Automatically? Please describe.

We have added a more detailed description in the legend of Figure 1 and we have revised the legend of Supplementary Figure 1 for clarity. In short, all routinely stained sections (both hematoxylin & eosin [H&E] and immunohistochemistry) were retrieved, revisited by light microscopy, and manually selected to represent a panoramic central slice of each metastasis in each patient. Slide selection was based on microscopic morphology, the description of the slides' origin in the standardized pathology reports, and done with guidance from the

macroscopic photo linked to the pathology report (as shown in Supplementary Figure 1). The individual slides were then digitized and manually annotated.

• What was the rationale behind “only” measuring length of a certain growth pattern? One can imagine that a lot of information / accuracy might be lost due to this 1-dimensional measurement. Have the authors tried using an automated (AI / ML based) approach of detecting the different growth patterns? QuPath as well as other programs could be used to train a classifier (desmoplastic / tumor / liver) which could then be utilized to determine the predominant growth pattern. This would be more objective, less error prone and could easily be automated even for large numbers of WSIs.

We agree with the reviewer that machine-learning tools could significantly expand the data that can be extracted from H&E images in general. Regarding liver metastases, we have discussed the recent advances in this field in the current clinical scoring guidelines (Latacz et al., Br J Cancer, 2022, PMID 35650276), where we present two machine-learning tools under development by us and others to extract growth pattern information. However, in our hands (Latacz et al.), these tools are still far from the accuracy of a human scorer. The main reasons for their limited performance are two-fold: (1) growth pattern scoring takes into account spatial information of the microenvironment, such as the presence of large portal triads in the vicinity of the invasion front, where a growth pattern is not scored, which as of yet is difficult to implement in automated algorithms, and (2) challenges in generating a robust model that reliably handles inter-slide variation induced by, e.g. staining intensity differences.

In the future, automated scoring will hopefully shed light on which aspects of the growth patterns (e.g., liver-tumor contact, perimetastatic inflammation, or numbers/location of incorporated liver parenchymal remnants) are prognostically most important.

Even when automated scoring will become available, a 3-dimensional assessment would require complete histological sampling and sectioning of each metastatic nodule, regardless of its size. Although our pathology department performs a rather extensive sampling regarding general clinical standards, this is far from a complete sampling and sectioning for histology of every metastatic nodule, which is due to the enormous resources this would require. However, some of us are part of the Belgium-led POEM study (Prospective complete histopathological characterization of liver metastases from colorectal and breast carcinoma to predict the histopathological growth patterns by medical imaging), in which complete metastases are sampled. Nevertheless, POEM will not be able to perform complete sectioning and reconstruction of the nodules (which would result in tens of thousands of standard slides), while we hope that this can be done for a limited number of smaller metastases in the future.

• The authors state that “[the] Boolean model [...] is difficult to reconcile with the probabilistic character of biological systems” (line 136 f) and argue that their approach better addresses the likely continuous nature of these changes. However, as far as I understand, all they have done is introduce another cut point at 33%. How were these thresholds / strata / cut points determined? Have the authors tried statistical methods to identify the optimal cut points (i. e. via ROC analysis or capabilities from the survival / survminer package in R, etc.)? I would be really interested in the Kaplan-Meier-curves, when 5 or more strata (0-20%, 21-40%, 41-60%, etc.) of desmoplastic growth are used. As there are two main groups (desmoplastic vs. replacement), this would mean that for the replacement type the correlation would be exactly inverse. Is this the case?

The reviewer raises an important point. First, to benchmark our extended scoring, we have utilized the same strata as those previously suggested when it was argued that only fully

encapsulated metastases define a group with superior prognosis (first paper to make this claim: Galjart et al., *Angiogenesis*, 2019. PMID 30637550; recently used in the clinical scoring guidelines on a large, multicenter cohort: Latacz et al., 2022). We appreciate the reviewer's suggestions for improving the clarity of our reasoning to define the strata, and we have incorporated the previously used strata (Latacz et al.) to investigate our data as Supplementary Figure 3a, which now displays five strata for desmoplastic encapsulation. A comparison of the largest published dataset using visual scorings to our data is included as **Reviewer Figure 1** below.

Reviewer Figure 1: Comparison of overall survival using the same strata as in the current scoring guidelines. a) Overall survival according to the strata indicated with different colors in Latacz et al., PMID35650276. Note that fully encapsulated tumors (“100% desmoplastic”) have the best outcome, and that the remaining curves do not follow the order of the degree of encapsulation (“33.1 – 67% non-desmoplastic” has better survival at 10 years compared to 0.1 – 33% “non-desmoplastic”). b) Overall survival using the same strata as in (a) in our cohort and using extended scoring. The right panel summarizes the data into three strata, which we propose is most illustrative to show the proportion-dependency of the outcome on fractions of encapsulation.

We have conducted a cut-point analysis using the maximally selected rank statistics on the encapsulated pattern, which determined the optimal cut-point for encapsulation to be 94.5%. This differs from the clinically suggested cut-point of 100%. However, such analyses can only reasonably define one cut-point, which, according to our line of argumentation, does not fully capture the proportion-dependency of the growth patterns. To address this aspect, we have included in the revised manuscript the results of the Cox-proportional hazard model for encapsulation in 0.1 fraction increments in the revised manuscript. This analysis identified the hazard ratio (HR) for death at 0.91 (95% confidence interval [CI] 0.87 – 0.95, $p=2e-04$, Wald test) and for liver-specific relapse (HR for hRFS) at 0.92 (95% CI: 0.88 – 0.96, $p=4e-05$, Wald

test), providing strong support for the proposed proportion-dependency (added in the first section of the Results).

Finally, we have included a Kaplan-Meier plot based on strata of replacement-type growth (Supplementary Figure 3b). These strata indeed mirror those of encapsulation, with higher fractions of replacement-type growth correlating with worse overall survival.

• The association of growth pattern with chemotherapeutic effects is interesting but needs to be explored in more detail. Clinically, the effect of chemotherapy on liver metastasis is usually scored using Rubbia-Brandt et al. (PMID: 17060484) for example. Have the authors data on the distribution of the growth pattern with respect to this response assessment? How do the authors explain that there is also a correlation between desmo GP and viable tumor cells in the chemonaïve subgroup?

We concur with both this reviewer and reviewer #2 that the association between chemotherapy and encapsulation is intriguing and warrants further investigation. To address this point, we have incorporated data from in vivo experiments demonstrating that chemotherapy and tumor cell ablation induce encapsulation, which we discuss in greater detail below in response to reviewer #2.

We have conducted a comprehensive assessment of the response to chemotherapy in our cohort, utilizing the quantitative approach by Blazer et al. (PMID: 18936472). This method is based on the percentage of residual tumor cells in relation to the total metastasis area and is the clinical standard in our pathology department. As such, we have documented the percentages of residual tumor cells for every slide in the study, which can be seen in Figure 6c and d. We have favored the approach by Blazer et al. because 1), it yields quantitative data that is best suitable for correlation analysis with the growth pattern percentages; 2) this system is routinely employed by our pathologists in clinical practice, which ensures the robustness and reproducibility of the derived data; 3) the tumor regression grade (TRG) according to Rubbia-Brandt considers fibrosis as a central pillar in the assessment and therefore confounds the investigation of associations between tumor “viability”/regression and perimetastatic fibrosis, which is key in our investigations.

Our proposed model of growth pattern plasticity is that the capsule develops when the tumor's capacity for replacement-type growth declines. The model makes no claims on the type of tumor cell impairment that underlies decreased tumor cell fitness. Rather, a prediction is that capsule formation is agnostic to the type of injury. Similarly, a reduced ability for replacement growth in the chemonaïve subgroup may result from a defect in tumor cell fitness to adapt to and progress in the liver microenvironment, or from partial tumor regression due to insufficient vascular supply. In either scenario, the balance between the capacity for replacement growth and the reparative-injury liver response favors perimetastatic capsule formation.

• For the argument that the desmoplastic stroma originates from liver damage and has less to do with the tumor cells themselves, it would be important to compare colorectal metastases to that of other primary tumors. Do the authors have any data on this? For example, from clinical experience, there are tumor types with a lower rate of desmoplastic stroma and better prognosis (e. g. neuroendocrine liver mets) and tumor types with higher rate of desmoplastic stroma and a worse prognosis (e. g. pancreatic liver mets). With the lack of functional experiments, more data would be needed to draw this conclusion.

We appreciate the suggestion to include examples of metastases from other primary tumors, and we have now incorporated data for liver metastases from melanoma, breast cancer, pancreatic cancer, and gallbladder cancer, as well as from intrahepatic cholangiocarcinoma and hepatocellular carcinoma (Supplementary Figure 7). In all of these cases, encapsulated metastases were observed, and the capsule consistently contained liver remnants. Additionally, protein profiles across the capsule's diameter were similar to those seen in the large colorectal cancer cohort.

Encapsulation has previously been associated with better outcomes in metastases from various primary cancers, which we discuss in the introduction. Our discovery of histological similarities shows that capsule formation likely is a general process, irrespective of the primary tumor type. Therefore, they further support a model in which non-tumor-specific factors result in capsule formation.

• Another key factor to prove or disprove the author's theory would be to assess the function of the surrounding liver. It would therefore be very important to include data on the function / appearance / morphology of the normal adjacent liver tissue. With hundreds of patients included in the study, there should be a subgroup of patients which has some form of liver pathology (i. e. fatty liver disease, fibrosis, cirrhosis, etc.). How is the distribution of desmoplastic / replacement pattern in these patients? Also, it would be really interesting to see the distribution of these growth patterns in tumors originating from the liver themselves (i. e. HCC) and the effect on clinical outcome.

In response to the reviewer's suggestions, we have incorporated new data on the morphology of the surrounding liver:

- 1. We analyzed clinical scorings of non-tumorous liver pathology (including fibrosis, inflammatory activity, presence of Periodic acid–Schiff–diastase [PAS-D] globules, and parenchymal iron deposits) in the context of the growth patterns, which are now included as Supplementary Table 2. We did not detect significant differences in these parameters between replacement and encapsulated metastases. These findings support the model that we propose, which suggests that impaired tumor growth is the primary requirement for capsule development, while the status of the surrounding liver is less important.*
- 2. In an extension to our initial cohort, we have identified patients with an established diagnosis of liver fibrosis/cirrhosis prior to CRLM development. Out of more than 600 consecutive patients, we found three cases, in line with the overall low frequency of cirrhosis in Sweden (added as Supplementary Figure 9). Two of these patients had encapsulated metastases, while one had predominantly replacement-type growth. Upon examining the histology of the replacement case, we observed that tumor invasion into the liver parenchyma progresses despite bridging fibrosis due to cirrhosis. Our interpretation of these data is that a profibrotic state of the liver in some liver zones, as seen in various fibrotic conditions, does not necessarily preclude replacement-type invasion into the remaining bulk of hepatic parenchyma.*
- 3. Regarding primary liver tumors, the CRLM classification originates from initial observations made in hepatocellular carcinoma (HCC; Nakashima et al., Human Pathology 1982, PMID 6176525), which have established a survival advantage for encapsulated HCC. In the revised manuscript, we include representative images of the invasion front histomorphology for intrahepatic cholangiocarcinoma (desmoplastic: Supplementary Fig 7c, replacement-type: Supplementary Fig 9b) and HCC (desmoplastic: Supplementary Figure 7e).*

- **For the quantification of the marker expression (curves) please provide the respective standard deviations.**

We appreciate the suggestion and have added an error metric to all marker expression plots.

- **The methodology of determining different types of portal triad destruction should be explained in better detail and supported by either data or literature. It is usually dependent on the type of liver damage and does not always follow the sequence vein, bile duct, artery. Please comment.**

We agree that the mechanisms behind portal triad atrophy and destruction can vary depending on the underlying cause. However, for the purpose of our study, the crucial comparison is between double-conserved portal triads (artery + bile duct) and single-conserved portal triads (artery alone). We argue that a remnant consisting of only an artery generally represents a more degraded status than a remnant that consists of both an artery and a bile duct. The proposed sequence of portal triad atrophy (portal vein, bile duct, artery) in the fibrous rim and central region of liver metastases is supported by the observation of portal tract remnants composed of artery + bile duct (A+BD) or artery alone (A), both with a consistent absence of the portal vein. This indicates that the portal vein is the first structure damaged in the process of portal triad atrophy in liver metastases.

The concept of porto-sinusoidal vascular disorder (PSVD) has recently emerged as a distinct pathological entity characterized by the absence of cirrhosis and the presence of histological findings such as obliterative portal venopathy/portal vein stenosis and vascular portal tract anomalies (Gottardi et al., J Hepatol, PMID 35690264). PSVD changes can present concomitantly with other liver diseases such as alcohol-induced or viral hepatitis. The pattern of portal zone atrophy described in our study bears a striking resemblance to portal vein stenosis described in PSVD. Upon revising the slides and in light of the histological findings described by Gottardi et al., we have updated Supplementary Figure 6c (former Supplementary Figure 2b) by reclassifying the distorted vascular channels, which we consider to be consistent with abnormal/remodeled portal/periportal vessels, rather than “preserved branches of the portal vein” as in the previous manuscript version. We have added this reference to the manuscript.

Another implication of the data is that liver metastases are mainly perfused by branches of the hepatic artery rather than through the portal vein. This is well-aligned with the literature on imaging of liver metastases, which supports that CRLM are perfused predominantly by the hepatic artery (Archer, Br J Surgery 1989, PMID 2474356).

Discussion:

- **Currently, the discussion is more of a repetition of the results section and needs to be revised. What is novel about the findings? What are the limitations / open questions? Etc.? To me it is unusual to refer to the figures in the discussion as well.**

We have revised the entire discussion and removed repetitive statements and references to the Figures.

- **The passage starting from line 409 to line 415 is somehow speculative as the authors have not provided any functional experiments in this regard. This should be framed differently.**

We have provided functional data from in vivo experiments to support the first claim of this statement “..our data support a model in which the desmoplastic rim primarily develops largely independent of active juxtacrine or paracrine signals from tumor cells, but rather as the combined result of liver injury and failed replacement-type growth”. In addition, we have removed the speculative second part of this passage, in which we discussed that replacement-type growth represents initial tumor stages.

Reviewer #2 (Remarks to the Author): expertise in bioinformatics of digital pathology

This is an interesting study addressing a clinically and biologically relevant topic. This study is the result of a lot of work, which the authors should be commended on. This involves a lot of work in sample collection, slide preparation, annotation and analysis.

We thank the reviewer for valuable input on our manuscript.

However, there are two major flaws in the scientific reasoning.

The first major issue is the claim of causality. The whole study is descriptive and the authors cannot deduct that the desmoplastic reaction is caused by the liver instead of the tumor. This is something that you really need a controlled experiment for, typically a mouse experiment. The authors present an impressive amount of descriptive data on the morphology of the desmoplastic rim, and about gradients of molecules in the desmoplastic rim. But all of these are still descriptive and have several possible reasons - they are not proof of causality. Even Figure 6 is compatible with the tumor tissue inducing scarification of the surrounding liver tissue, as opposed to scarification induced by peritumoral liver tissue. The authors could fix this issue by removing claims of causality from this descriptive study, but this reduces the novelty.

We agree that the original study was descriptive in nature. However, we do not intend to claim that the liver causes the desmoplastic rim per se. Based on a spatial analysis of cellular constituents and protein gradients, we infer that the rim represents a hepatic injury response to the insult of an invading tumor. We hypothesized that the rim might develop when the balance between tumor invasion and reparative liver processes favors the latter. We have restructured the manuscript to more accurately reflect this line of reasoning, which previously has been unclear.

We acknowledge that functional data to support our hypothesis were lacking. In the revised manuscript, we have incorporated in vivo experiments to address these valid points (Figure 7 and Supplementary Figure 10).

First, we generated liver metastases using the colorectal cancer cell line, MC38 (see also the suggestion by reviewer #3). We treated mice bearing MC38 liver metastases with chemotherapy and observed an increased degree of encapsulation, albeit at low overall frequencies. Next, we employed KPC-T cells (a pancreatic cancer cell line) to generate liver metastases. We engineered the KPC-T cells to express the diphtheria toxin receptor, enabling us to ablate established metastases with diphtheria toxin. Tumor cell ablation led to increased encapsulation, and tumor cell viability was strongly negatively correlated with encapsulation (Figure 7 and Supplementary Figure 10).

These functional experiments establish, for the first time, that the capsule forms when tumor cell fitness is impaired.

The second major issue is the presence of selection biases in the cohort, which casts doubt on the claim that chemotherapy induces desmoplasia. The authors find that patients after neoadjuvant chemotherapy have more desmoplastic growth than chemo-naïve patients. However, this is highly biased due to two types of selection bias. (1) Oncologists and surgeons may be more likely to prescribe neoadjuvant therapy if the tumor seems slowly growing, as opposed to "out of control", even at the same absolute size of tumors. Hence, more favorable biological patterns can be enriched in the population which receives neoadjuvant therapy. (2) Even more gravely, neoadjuvant therapy selects the patient population for good prognosis patients. Patients who progress under therapy are less likely to be operated. Hence, the survivors of 2-3 months of chemo are enriched for favorable-biology tumors, i.e. desmoplastic mets. The authors discuss this in line 245-250, but I do not find that this argument resolves the bias.

We concur with the reviewer's valid point regarding the potential skew of the neoadjuvant group towards the encapsulated phenotype, which is due to the exclusion of patients who progress during neoadjuvant treatment, as it is usual in clinical routine. However, we would like to point out that this selection bias specifically affects the analysis of the effect of chemotherapy.

By adding the in vivo data, we have provided functional evidence that chemotherapy can induce encapsulation. Furthermore, to approximate the impact of selection bias for the analysis of the chemotherapy effect, we performed additional analyses and extended our initial cohort:

- 1. We have added data on patients that were operated on despite progression on chemotherapy, collected from an extended patient cohort of more than 600 patients with colorectal liver metastases (Figure 6f, Supplementary Figure 8b). We found that none of the progressors (n=9 over a 9-year period) had predominantly encapsulated metastases, which supports our interpretation that the encapsulation represents failed tumor invasion, while replacement represents actively growing, aggressive metastases.*
- 2. We used these data to estimate the impact of the potential selection bias after chemotherapy. First, we defined the relative frequency of progressors on chemotherapy by reviewing all the tumor board decisions and radiological examinations in the Stockholm region for patients with colorectal cancer liver metastases in a 5-year period. After having defined the frequency of progressors on chemotherapy, we injected virtual progressors matching that frequency in our cohort to the comparison of the growth patterns and observed that the difference remained significant with a higher degree of encapsulation (Supplementary Figure 8c).*

As the reviewer points out, CRLM patients are selected for operation and to receive neoadjuvant chemotherapy. Unfortunately, these clinical decisions are not made based on well-defined clinical parameters but individually, factoring in various clinical factors on multidisciplinary tumor boards. However, in most patients, there is no data available on the rate of tumor progression at the time a decision on preoperative chemotherapy is made, so this metric is not factored in when deciding on neoadjuvant treatment. Nevertheless, decisions for and against chemotherapy are not made randomly, and, hence, significant biases cannot be excluded. However, a recent study, which we cited (Nierop et al., J Pathol Clin Res. 2022 Jan) has analyzed patient samples from the key randomized-controlled clinical trial that has led to clinical implementation of neoadjuvant chemotherapy in CRLM (Nordlinger et al., The Lancet, 2013 (24120480). They compared the growth patterns between patients randomly assigned to

either surgery or neoadjuvant chemotherapy followed by surgery, and they found increased encapsulation after chemotherapy (Nierop et al., 2022). Our in vivo data provide first functional evidence for this clinical observation.

In the revised manuscript, we discuss more prominently the previously published analysis of the growth patterns in a randomized controlled trial (Discussion).

Minor issues:

The authors mention "60k annotations" in their abstract, but this information is not helpful. What the reader is interested in is the number of tissue samples, and the number of patients. You can even generate 60k annotations in a single patient. Please state the patient number in the abstract.

We have amended the abstract accordingly.

The study should adhere to the STARD/TRIPOD or other relevant guidelines by the Equator network.

We appreciate the suggestions. The human data are reported according to REMARK, as we felt this reporting guideline was most suitable, even though our study is not investigating a new biomarker. The animal experiments are reported according to ARRIVE. Checklists have been submitted to the journal along with the submission of the revision.

Reviewer #3 (Remarks to the Author): expertise in molecular biology of colorectal cancer liver metastasis

The manuscript by Moro et al investigates the formation of two distinct type of colorectal cancer (CRC) liver metastases, which are referred to as desmoplastic and replacement type lesions. Patients with desmoplastic CRC liver metastases have better outcomes when compared to those with replacement type lesions. Previous studies suggested that only patients with 100% desmoplastic features exhibited better outcomes; however, if any portion of the metastases were replacement, the outcome was worse. The current manuscript argues that the proportion of desmoplastic features has prognostic significance. The authors perform detailed histological characterization of the desmoplastic ring from patient-derived CRC liver metastasis samples and put forth a model in which the stromal deposition is the result of an injury response in the liver, that increases deposition of the encapsulating stroma, rather than tumor-induced activation of cancer-associated fibroblasts/stellate cells in the liver.

The authors are investigating an important question regarding the formation of desmoplastic CRC liver metastases, which is an understudied and important area of research. The data presented in the manuscript comes from a careful characterization of desmoplastic lesions using an excellent resource of clinical material obtained from patients with CRC liver metastases. the authors are to be commended for the efforts invested to assess the degree of the desmoplastic ring around the entire circumference of the metastatic lesion. While the authors interpretation that desmoplastic lesions form as the result of a liver injury response that encapsulates the lesion and prevents parenchymal invasion, it would be important for the authors to test some of the

predictions they make using pre-clinical mouse models. These are discussed in detail below.

We would like to thank the reviewer for the thorough assessment of our manuscript and helpful suggestions.

Major comments:

1) The authors will need to carefully review the figure legends to ensure that the figures are labeled correctly. See notes under minor comments for specific examples of the disconnect between the text and the figure legends.

We sincerely apologize for this oversight.

2) The authors argue that the cells present at the desmoplastic rim/liver interface are more similar to activated fibroblasts that are seen in the context of a wound-healing, fibrotic response. In contrast, the fibroblasts located close to the desmoplastic rim/cancer interface would be more reminiscent of cancer-associated fibroblast phenotype. It would strengthen the paper considerably if the authors could generate some additional data that would more clearly define these distinct fibroblast cell types. This could include spatial transcriptomic approaches or laser capture microdissection of alphaSMA+/NGFR+ cells (outer portion of the desmoplastic ring – adjacent to the liver) versus alphaSMA+/NGFR- cells (inner portion of the desmoplastic ring – adjacent to the metastatic lesion). Alternatively, could fibroblasts/stellate cells be isolated from patient material and separated into NGFR+ and NGFR- cells for RNA sequencing.

We agree that a comprehensive characterization of the capsule and its distinct fibroblast cell types would be valuable to further understand its development and biological implications. Some of us are part of several efforts to achieve an unsupervised characterization of the capsule, including a spatial transcriptomics (ST) approach that has recently been published (Fleischer et al., Mol Cancer 2023, PMID 36691028); using ST we found that the resolution of ST techniques, which currently is around 100 μm on formalin-fixed paraffin-embedded tissues, is insufficient to resolve the zonation of the capsule.

We appreciate the suggestion to use NGFR as a surface marker for sorting, which could give valuable insights. An important caveat could be that enriching for NGFR⁺ cells is sensitive to the contamination from non-capsule NGFR⁺ cells such as periportal fibroblasts, which requires spatial techniques for carefully characterizing the rim.

To further characterize the capsule at the required resolution, and considering feasibility within the time constraints of this revision, we have added two sets of data:

- 1. Multiplex-immunofluorescence with a panel of stromal markers (included as Figure 3d-f, Supplementary Figure 5 a-b): The results identify the outer part of the capsule as ASMA^{high}/PDGFRa^{high}, while FAP dominates its inner part, providing additional stromal markers for future studies.*
- 2. HiPlex RNA in situ hybridization for 6 stromal markers (Supplementary Figure 5c-d): The results confirm a zonation of the capsule and provide additional stromal transcripts for its characterization.*

Finally, we have added a Figure summarizing the spatial profiles of all the relevant markers that we have quantified in this study (Figure 5e).

3) The authors argue that a tissue response that leads to fibrosis might affect the type of

lesion that develops in patients with CRC liver metastases. If this is the case, would the authors expect that patients with a previous history or current condition associated with liver fibrosis (Cirrhosis) would be more susceptible to the development of desmoplastic lesions? Has such an association been investigated? Could the current patient cohort developed by the authors be used to investigate this?

We have added an analysis of underlying profibrotic liver conditions including assessments of fibrosis stages and grades of inflammation in the overall cohort. In addition, we have identified and analyzed patients with pre-existing, clinically established cirrhosis, as outlined in more detail in our answer to reviewer #1. The results suggest that a fibrotic state of the liver is, per se, insufficient for capsule development and that replacement growth is possible in the remaining parenchymal liver regions.

Our model did not predict that pre-existing liver fibrosis is sufficient for shifting the growth pattern to encapsulation. Rather, the data support a model, in which impaired tumor invasion is required for the maturing capsule to build. We have amended the discussion to clarify this model.

4) The authors suggest that the replacement type CRC liver metastases are the “default” type of lesion, and that desmoplastic lesions form secondary to a tissue repair/fibrosis response that encapsulates the lesion. Could this be tested in an animal model? Several CRC cell lines exist that form replacement type metastases when splenically injected (MC38 cells are an example). How would the formation of MC38-derived liver metastases be altered if these cells were injected into mice with active liver fibrosis (PMID: 34028753)? Would the default replacement type lesion normally formed by MC38 cells in normal liver be shifted to a desmoplastic lesion when injected into a fibrotic liver? Would you see the same distribution of NGFR+/alphaSMA+ and NGFR-/alphaSMA+ fibroblasts/stellate cells in such a model?

We thank the reviewer for this suggestion. We have added in vivo data from two experimental approaches (chemotherapy and diphtheria-toxin mediated tumor cell ablation) as outlined in more detail in response to reviewer #2.

As the reviewer predicted, replacement was the main mode of growth of MC38 metastases and of metastases from cells derived from a canonical pancreatic cancer model, KPCT. Chemotherapy and tumor cell ablation with diphtheria toxin led to the development of an ASMA⁺ capsule, connecting reduced tumor cell viability to capsule formation (Figure 7).

The suggested further functional experiments are relevant for understanding the role of the capsule in preventing replacement-type invasion of a tumor that is capable of replacement-type growth (such as MC38) in a fibrotic liver. However, our clinical data demonstrate no correlation of (pro-)fibrotic conditions and capsule formation, which goes in line with the concept that impaired colonization of the liver plates locally induces the fibrotic capsule, as we put forward in the discussion. Another general challenge with the suggested experiments could be to differentiate whether tumor cells merely align with the fibrous stroma in a cirrhotic liver, as appears to be the case in human replacement-type metastases in cirrhotic livers (Supplementary Figure 9). We believe that the ablation model that we have developed (based on KPCT-DTR cells) in the future could be a particularly valuable tool in this context, as it allows the ablation of an established tumor (Figure 7h), which could then serve as a tumor-induced fibrotic bed for a second ultrasound-guided injection of tumor cells, creating this situation.

It is unusual to reference a manuscript figure in the introduction. Maybe a paper citation can be used when describing the features of the desmoplastic and replacement lesions in the introduction and the figure can be cited in the results.

We have reorganized the text and referred to Figure 1a in the first part of the results section.

Care should be taken with the Figure legends:

Page 13 (PDF): Figure Legend should indicate Figure 2 (not Figure 1)

Page 14 (PDF): Figure Legend should indicate Figure 3 (not Figure 2)

Page 19 (PDF): Figure Legend should indicate Figure 4 (not Figure 3)

Page 22 (PDF): Figure Legend should indicate Figure 5 (not Figure 4)

Page 24 (PDF): Figure Legend should indicate Figure 6 (not Figure 5)

Page 28 (PDF): Figure Legend should indicate Figure 7 (not Figure 6)

Page 17/18 (PDF): References to Supplementary Fig. 3 in the text should be referring to Supplementary Fig. 4

We apologize for these errors.

REVIEWERS' COMMENTS

Reviewer #1 (Remarks to the Author):

The authors have addressed all of my concerns to the fullest and far exceeded what I would have deemed necessary for their manuscript to be published. They should be strongly commended for all the additional effort and work they put into this study. I have no further comments and can only recommend publication.

Reviewer #2 (Remarks to the Author):

The authors have improved the study by performing additional experiments. Thank you! No further requests.

Reviewer #3 (Remarks to the Author):

The revised manuscript by Moro et al. has been substantially improved upon revision, and the authors should be commended for a thorough response to all the reviewer's comments. With respect to the issues raised during my review, the authors have added additional multiplex IHF and in situ hybridization data to better define the fibroblast subtypes that exist at the fibrotic rim/liver interface versus the fibrotic rim/metastasis interface. Importantly, the authors have tried to identify a subset of patients that possessed underlying fibrotic processes prior to the detection of colorectal cancer liver metastases. Out of 600 patients analyzed, only 3 had liver fibrosis/cirrhosis. Of these, two presented with desmoplastic liver metastases and 1 possessed a replacement type lesion. Although these are very small numbers, the interpretation put forth by the authors that an underlying fibrotic condition is not sufficient to drive encapsulation around metastatic lesions is reasonable. Finally, the authors have included two in vivo animal models demonstrating that interventions that kill colorectal cancer or pancreatic cancer cells within established liver metastases is enough to trigger the formation of a fibrotic capsule. The authors interpret this data to mean that interventions that reduce cancer cell viability or fitness can lead to encapsulation. This idea differs slightly from the concept that was proposed in the first submission, where the authors argued that lack of aggressive cell invasion into the liver parenchyma was the trigger for a fibrotic response and lesion encapsulation. As a result of this new data, the authors are encouraged to harmonize this concept throughout the manuscript. Considering the extensive revisions performed by the authors, I am supportive of publication.

RESPONSE TO REVIEWERS' COMMENTS

We would like to thank the reviewers once again for their constructive feedback on our manuscript “*An idiosyncratic zonated stroma encapsulates desmoplastic liver metastases and originates from injured liver*”.

Based on the suggestion made by reviewer #3, we have reconsidered all the text passages related to the model of encapsulation that we propose based on our data and we have highlighted all those passages in the text of the edited manuscript. Below, we provide a point-by-point response to the remaining comments:

Reviewer #1 (Remarks to the Author):

The authors have addressed all of my concerns to the fullest and far exceeded what I would have deemed necessary for their manuscript to be published. They should be strongly commended for all the additional effort and work they put into this study. I have no further comments and can only recommend publication.

We thank the reviewer for the valuable and constructive comments that have significantly improved the analysis of our clinical cohort.

Reviewer #2 (Remarks to the Author):

The authors have improved the study by performing additional experiments. Thank you! No further requests.

We thank the reviewer for the comments that helped us improve the concept of our study and develop relevant mouse models.

Reviewer #3 (Remarks to the Author):

The revised manuscript by Moro et al. has been substantially improved upon revision, and the authors should be commended for a thorough response to all the reviewer's comments. With respect to the issues raised during my review, the authors have added additional multiplex IHF and in situ hybridization data to better define the fibroblast subtypes that exist at the fibrotic rim/liver interface versus the fibrotic rim/metastasis interface. Importantly, the authors have tried to identify a subset of patients that possessed underlying fibrotic processes prior to the detection of colorectal cancer liver metastases. Out of 600 patients analyzed, only 3 had liver fibrosis/cirrhosis. Of these, two presented with desmoplastic liver metastases and 1 possessed a replacement type lesion. Although these are very small numbers, the interpretation put forth by the authors that an underlying fibrotic condition is not sufficient to drive encapsulation around metastatic lesions is reasonable. Finally, the authors have included two in vivo animal models demonstrating that interventions that kill colorectal cancer or pancreatic cancer cells within established liver metastases is enough to trigger the formation of a fibrotic capsule.

We thank the reviewer for their appreciation of the additional data.

The authors interpret this data to mean that interventions that reduce cancer cell viability or fitness can lead to encapsulation. This idea differs slightly from the concept that was proposed in the first submission, where the authors argued that lack of aggressive cell invasion into the liver parenchyma was the trigger for a fibrotic response and lesion encapsulation. As a result

of this new data, the authors are encouraged to harmonize this concept throughout the manuscript.

Thank you for identifying the subtle yet important differences in the articulation of our model. We have highlighted in yellow all sections in the manuscript pertaining to this model to assess the descriptions throughout the manuscript. In the original manuscript, our emphasis was on the broad concept that impaired tumor cell ability to infiltrate the liver plates leads to capsule formation.

In the revised manuscript, to functionally validate this concept, we utilized two mouse models, where we demonstrated that compromising cancer cells, either through chemotherapy or genetic ablation, induces encapsulation. This shifts the focus to the *induced* impairment of invasion, rather than tumor-intrinsic differences in the ability to infiltrate the liver parenchyma. However, we argue that the observation of encapsulation in patients who have not received chemotherapy supports the role of inherent tumor cell-intrinsic factors, which aligns more closely with the initial wording.

We appreciate the suggestion reconcile the concept. Given the reasoning above, we have edited the text slightly, for example by adding in ll. 435 (“... rather, reduced tumor cell ability to colonize the liver plates, *for example as a result of chemotherapy*, emerged as a driver for capsule formation.”) and ll. 536 by adding the sentence “*Detailed studies on chemo-naïve, encapsulated metastases could unveil genetic or epigenetic features of cancer cells that fail at continuous replacement-type growth, leading to capsule development.*”

We hope that these and further small change better illustrate our model that both intrinsic tumor-cell factors and external influences can contribute to a reduction in tumor cell fitness, subsequently leading to encapsulation.

Considering the extensive revisions performed by the authors, I am supportive of publication.

Thank you!